

# Implementation of the sectional aerosol module SALSA into the PALM model system 6.0: Model development and first evaluation

Mona Kurppa[1], Antti Hellsten[2], Pontus Roldin[1,3], Harri Kokkola[4], Juha Tonttila[4], Mikko Auvinen[1,2], Christoph Kent[5], Prashant Kumar[6], Björn Maronga[7,8], and Leena Järvi[1,9]

[1]Institute for Atmospheric and Earth System Research / Physics, Faculty of Science, University of Helsinki, P.O. Box 68, 00014 Helsinki, Finland
[2]Finnish Meteorological Institute, 00101 Helsinki, Finland
[3]Division of Nuclear Physics, Lund University, 22100 Lund, Sweden
[4]Finnish Meteorological Institute, 70211 Kuopio, Finland
[5]Department of Meteorology, University of Reading, Reading RG6 6BB, United Kingdom
[6]Global Centre for Clean Air Research (GCARE), Department of Civil & Environmental Engineering, University of Surrey, Guildford GU2 7XH, United Kingdom
[7]Leibniz University Hannover, Institute of Meteorology and Climatology, 30419 Hannover, Germany
[8]Geophysical Institute, University of Bergen, 5020 Bergen, Norway
[9]Helsinki Institute of Sustainability Science, University of Helsinki, 00014 Helsinki, Finland

*Correspondence to:* Mona Kurppa (mona.kurppa@helsinki.fi)

**Abstract.** Urban pedestrian-level air quality is a result of an interplay between turbulent dispersion conditions, background concentrations and heterogeneous local emissions of air pollutant and their transformation processes. Still, the complexity of these interactions cannot be resolved by the commonly used air quality models. By embedding the sectional aerosol module SALSA to the large-eddy simulation model PALM, a novel, high-resolution, urban aerosol modelling framework has been developed. The first model evaluation study on the vertical variation of aerosol number concentration and size distribution in a simple street canyon without vegetation in Cambridge, UK, shows excellent agreement with measurements. Dispersion conditions and local emissions govern the pedestrian-level aerosol number concentrations. Out of different aerosol processes, dry deposition is shown to decrease the total number concentration by over 20 %, while condensation and dissolutional increase the total mass by over 10 %. Following the model development, the application of PALM can be extended to local- and neighbourhood-scale air pollution and aerosol studies that require a detailed solution of the ambient flow field.

# 1 Introduction

The coincidence of rising population densities, high air pollutant emissions and limited ventilation in urban areas leads to an increasing number of air pollution related health problems and premature deaths globally every year (Gakidou et al., 2017; WHO, 2016). The local air quality is an outcome of complex interactions between the urban landscape, meteorology, back-



ground pollutant concentrations and local emissions as well as chemical and physical processes of air pollutants. Thereby, urban air pollutant concentration fields are highly irregular in both time and space (e.g. Kumar et al., 2011). At the same time, pollutant characteristics such as the size of aerosol particles and chemical compositions of both particles and gaseous mixtures are essential factors in determining the health impacts (for review, see e.g. Kelly and Fussell, 2012). Traditionally used local

urban air quality models, such as Gaussian dispersion or semi-empirical street pollution models, cannot resolve these details in concentration fields and interactions due to an inadequate representation of the urban complexity.

    Detailed information on the air pollutant concentration variability in urban areas is however highly valuable to urban planning in order to design healthy living environments (Giles-Corti et al., 2016; Kurppa et al., 2018), to air quality monitoring network design and exposure studies. Therefore, a building-resolving tool for simulating and predicting air quality in real complex

urban environments in current and future conditions is needed. To determine the air flow and dispersion, computational fluid dynamics (CFD) models and notably the large-eddy simulation (LES) are currently the most promising methods. Compared to LES, Reynolds-averaged Navies-Stokes (RANS) -based turbulence models can be computationally less demanding but their ability to resolve the instantaneous turbulence structures above a complex urban surface is shown to clearly underperform (e.g. Antoniou et al., 2017; García-Sánchez et al., 2018, and references within). By either method, the computational costs have

been the bottleneck in extending CFD based air quality modelling from tail-pipe emission studies (e.g. Huang et al., 2014; Liu et al., 2011) to the neighbourhood scale studies. Currently, there exists a number of RANS and LES models coupled with some chemical mechanism (Zhong et al., 2016), a few RANS models with an aerosol module, for instance Mercure_Saturne with MAM (Albriet et al., 2010) and ANSYS Fluent based models (Uhrner et al., 2007; Huang et al., 2014) such as CTAG (Wang and Zhang, 2012), and at least one LES model including a detailed aerosol module (Liu et al., 2011), which is however only

applied in a tail-pipe emission study. The CTAG model has also been run in a LES mode (Steffens et al., 2013), but to date only aerosol dry deposition has been studied (Tong et al., 2016a, b) and the chemical composition has been usually ignored.

    The fate of aerosol particles in the atmosphere depends substantially on their size distribution. The numerical approaches to describe the aerosol size distribution and to solve the aerosol general dynamic equations can generally be divided into modal, moment and sectional approaches. The modal aerosol modules (Ackermann et al., 1998; Liu et al., 2012; Vignati et al.,

2004) represent the continuous aerosol size distribution as an superposition of several modes (usually log-normal distributions), whereas moment-based methods track the lower-order radial moments of the aerosol size distribution (McGraw, 1997). Both approaches are computationally efficient due to the small number of prognostic variables. However, the modal approach lacks accuracy in simulating the evolution of the aerosol size distribution, especially if the standard deviations of log-normal modes are not allowed to vary (Whitby and McMurry, 1997; Zhang et al., 1999). Applying the moment approach, instead, requires

resolving a closure problem of the moment evolution equations (Wright et al., 2001). Furthermore, as aerosol properties are tied into moments, which are typically not observed properties except for the first moments, retrieving information on aerosol properties during the simulation increases the computational load. In the sectional approach (Gong et al., 2003; Zaveri et al., 2008; Zhang et al., 2004), the aerosol size distribution is represented as a discrete set of size bins. Sectional approach is flexible and accurate, but usually computationally more demanding due to the high number of prognostic variables.



To meet the needs of a high-resolution urban air quality model that can account for the complex interactions controlling the local air quality at the neighbourhood to city scale, this article presents the implementation of the aerosol module SALSA (Sectional Aerosol Module for Large Scale Applications, Kokkola et al., 2008) as a part of the PALM model system (see Maronga et al. (2015) for the description of PALM 4.0; a description of version 6.0 is envisaged in Maronga in this special

issue of Geoscientific Model Development). The aim is to include the aerosol dynamic processes into PALM, evaluate the model performance under different meteorological conditions, and study the relative impact of aerosol processes on the aerosol size distributions and chemical compositions in real urban environment.

The modelling methods and equations of SALSA, implementation to PALM, computational costs and inevitable numerical issues related to the sectional representation are discussed in Sect. 2. The model evaluation set-up and sensitivity tests are

described in Sect. 3 and the model results in Sect. 4. Finally, Sect. 5 discusses the applications and limitations of the model.

## 2  Model description

### 2.1  PALM

The PALM model system (version 6.0) features an LES core for atmospheric and oceanic boundary layer flows, which solves the non-hydrostatic, filtered, incompressible Navier-Stokes equations of wind ($u, v$, and $w$) and scalar variables (sub-grid

scale turbulent kinetic energy $e$, potential temperature $\theta$ and specific humidity $q$) in Boussinesq-approximated form. Note that PALM, originally developed as a pure LES code, nowadays also offers a RANS-type turbulence parametrisation. PALM is especially suitable for complex urban areas owing to its features such as a Cartesian topography scheme, a plant canopy module and recent model enhancements as so-called PALM-4U (short for: PALM for urban applications) components such as an urban surface scheme (first version described in Resler et al., 2017) and a land surface scheme (first description in Maronga

and Bosveld, 2017). Furthermore, other PALM-4U components, such as chemistry and indoor climate modules, have or are currently being implemented in the PALM model system to develop a modern and highly-efficient urban climate model. Due to its excellent scalability on massively parallel computer architectures, PALM is applicable for carrying out computationally expensive simulations over large, neighbourhood- and city-scale domains with a sufficiently high grid resolution for urban LES (Auvinen et al., 2017; Xie and Castro, 2006). The performance of PALM over urban-like surfaces has been successfully

evaluated against wind tunnel simulations, previous LES studies and field measurements (Kanda et al., 2013; Letzel et al., 2008; Park et al., 2015; Razak et al., 2013). Some fundamental technical specifications of PALM are represented in Table 1.

### 2.2  SALSA

SALSA was selected as the basis for representing aerosol dynamics in PALM since one major criteria in its development has been limiting computational expenses without the cost of accuracy. Despite being originally designed for global-scale

applications, SALSA is equally suitable for presenting aerosol dynamics also at local scale. In SALSA, the aerosol number size distribution is discretised into $X_{\mathrm{B}}$ size bins $i$ based on the mean dry particle diameter $\overline{D}_i$ of each bin. The number $n_i$ (m$^{-3}$) and





**Table 1.** The technical specifications of the LES model PALM.

| Property | Characteristics |
| --- | --- |
| Programming language | Fortran 95/2003 |
| Discretisation in space | Arakawa staggered C-grid (Harlow and Welch, 1965; Arakawa and Lamb, 1977) |
| Parallelization | Two-dimensional decomposition (Raasch and Schröter, 2001). Communication between processors realized using Message Passing Interface (MPI). Also OpenMP parallelization of loops and a hybrid mode allowed. |
| Sub-grid scale closure | 1.5-order scheme based on Deardorff (1980) and modified by Moeng and Wyngaard (1988) and Saiki et al. (2000). |
| Time-integration scheme | 3rd-order Runge-Kutta approximation (Williamson, 1980). |
| Wall model | By default Monin-Obukhov similarity theory (MOST, Monin and Obukhov, 1954). Is a surface scheme is switched on, the momentum flux is calculated via MOST, while surface fluxes of sensible and latent heat are calculated based on an energy balance solver for the surface temperature and a party MOST-based resistance parametrisation. |

mass concentration $m_{c,i}$ ($\mathrm{kg\,m^{-3}}$) of each chemical component $c$ are the model prognostic variables. SALSA was originally optimized for computationally expensive large scale climate models, and therefore, the number of size bins is kept to the minimum (default $X_{\mathrm{B}} = 10$) and only the following chemical components can currently be included: sulphuric acid ($H_2SO_4$), organic carbon (OC), black carbon (BC), nitric acid ($HNO_3$), ammonium ($NH_3$), sea salt, dust and water ($H_2O$). Furthermore,

the gaseous concentrations of $H_2SO_4$, $HNO_3$, $NH_3$ and semi- and non-volatile organics (SVOC and NVOC) are also default prognostic variables. Nitrates and ammonia were not included in the original SALSA but have been later added (Kudzotsa et al.). The sectional size distribution can be further divided into subranges 1 ($\overline{D}_i \lesssim 50$ nm) and 2 ($\overline{D}_i \gtrsim 50$ nm). Subrange 1 consists of smallest particles assumed to be internally mixed, strongly hygroscopic, and containing only $H_2SO_4$, OC, $HNO_3$ and/or $NH_3$. Subrange 2 can contain all chemical components and it can be further divided into strongly hygroscopic (2a) and

weakly hygroscopic (2b) subranges to allow for the description of externally mixed aerosol particle populations (Kokkola et al., 2018). The evolution of aerosol size distribution is represented using the sectional hybrid-bin method (Young, 1974; Chen and Lamb, 1994). As a difference to the original SALSA, $\overline{D}_i$ is calculated as the geometric mean diameter instead of the arithmetic mean. Assuming spherical particles, the later tends to overestimate the total volume $\overline{V}_i = \frac{\pi}{6}\overline{D}_i^3$ especially for larger aerosol particles when $X_{\mathrm{B}} \sim 10$.

The original SALSA contains detailed descriptions for the aerosol dynamic processes of nucleation, condensation, dissolutional growth and coagulation, and here it has been further extended by including dry deposition on solid surfaces and resolved-scale vegetation and gravitational settling. Resuspension of fine aerosol particles from surfaces is usually negligible



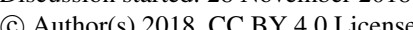


and hence neglected. Detailed description of the aerosol source/sink terms is given below (and in Kokkola et al. (2008) and Tonttila et al. (2017)).

### 2.2.1 Coagulation

Coagulation decreases the aerosol number as two aerosol particles collide to form one larger particle. In SALSA, coagulation is solved using the non-iterative method by Jacobson (2005). For $n_i$,

$$n_{i,t} = \frac{n_{i,t-\Delta t}}{1 + \Delta t \sum\limits_{j=i+1}^{X_B} \beta_{i,j} n_{j,t-\Delta t} + \frac{1}{2}\Delta t \beta_{i,i} n_{i,t-\Delta t}} \tag{1}$$

and similarly, for $m_{c,i}$

$$m_{c,i,t} = \frac{\rho_c \left( v_{c,i,t-\Delta t} + \Delta t \sum\limits_{j=1}^{i-1} \beta_{j,i} v_{c,j,t} n_{i,t-\Delta t} \right)}{1 + \Delta t \sum\limits_{j=i+1}^{X_B} \beta_{i,j} n_{j,t-\Delta t}} . \tag{2}$$

Here, $t$ and $t - \Delta t$ are the current and previous time steps, $\beta_{i,j}$ is the coagulation kernel ($\mathrm{m^3\,s^{-1}}$) of the colliding aerosol particles in size bins $i$ and $j$, and $\rho_c$ is the density of chemical component $c$. The coagulation kernel $\beta_{i,j} = E_{\mathrm{coal},i,j} K_{i,j}$ is the product of a collision kernel $K_{i,j}$ ($\mathrm{m^3\,s^{-1}}$) and a dimensionless coalescence efficiency $E_{\mathrm{coal},i,j}$. For aerosol particles smaller than 2 µm in radius, $E_{\mathrm{coal},i,j}$ can be approximated as unity (i.e. particles stick together) as the likelihood of bounce-off is low (Beard and Ochs, 1984). Brownian coagulation is assumed for aerosol particles, for which $K_{i,j}$ in the transition regime is calculated with the interpolation formula by Fuchs (1964),

$$K_{i,j} = \frac{4\pi(r_i + r_j)(\Gamma_{p,i} + \Gamma_{p,j})}{\dfrac{r_i + r_j}{r_i + r_j + \sqrt{\delta_i^2 + \delta_j^2}} + \dfrac{4(\Gamma_{p,i} + \Gamma_{p,j})}{\sqrt{v_{p,i}^2 + v_{p,j}^2}(r_i + r_j)}} , \tag{3}$$

where $r_i$ (m) is the particle radius, $\Gamma_{p,i}$ ($\mathrm{m^2\,s^{-1}}$) is the particle diffusion coefficient, $\delta_i$ (m) is the mean distance from the centre of the sphere reached by particles leaving the surface of the sphere and travelling a distance of particle mean free path, and $v_{p,i}$ ($\mathrm{m\,s^{-1}}$) is the thermal speed of a particle in air.

### 2.2.2 Condensation and dissolutional growth

Condensation of gases on an aerosol particle increases the particle volume and decreases the gas phase concentrations. For water vapour, $H_2SO_4$, NVOC and SVOC condensation is calculated applying the analytical predictor of condensation scheme (Jacobson, 2005), in which the vapour mole concentration $C_{c,t}$ at time step $t$ after condensation is first calculated as

$$C_{c,t} = \frac{C_{c,t-\Delta t} + \Delta t \sum\limits_{i=1}^{X_B} \left( k_{c,i,t-\Delta t} S'_{c,i,t-\Delta t} C_{c,s,i,t-\Delta t} \right)}{1 + \Delta t \sum\limits_{i=1}^{J} k_{c,i,t-\Delta t}} , \tag{4}$$





where $k_{c,i,t-\Delta t}$ is the particle volume-dependent mass-transfer coefficient $(\mathrm{s}^{-1})$ in size bin $i$ at the previous time step $t - \Delta t$, $S'_{c,i,t-\Delta t}$ is the equilibrium saturation ratio and $C_{c,s,i,t-\Delta t}$ is an uncorrected saturation vapour mole concentration $(\mathrm{mol\,m}^{-3})$ of the condensing gas $c$. The change in particle mole concentration $c_{c,i,t}$ in the aerosol size bin $i$ is then given by a formula

$$c_{c,i,t} = c_{c,s,i,t-\Delta t} + k_{c,i,t-\Delta t}\left(C_{c,t} - S'_{c,i,t-\Delta t}C_{c,s,i,t-\Delta t}\right), \tag{5}$$

which is then translated to aerosol number and mass concentrations. Condensation and evaporation of water vapour on aerosol particles would require a very short time step to avoid non-oscillatory solutions. The applied solution used in SALSA is described in Tonttila et al. (2017).

Furthermore, aerosol particles may further grow due to dissolutional growth when a gas transfers to a particle surface and dissolves in liquid water on the surface. This partitioning between the gaseous and particulate phases is solved for water vapour,

nitric acid and ammonia using the analytical predictor of dissolution (APD) scheme (Jacobson, 2005) in the following way. First, the vapour mole concentration $C_{c,t}$ after dissolutional growth at time step $t$ is calculated as

$$C_{c,t} = \frac{C_{c,t-\Delta t} + \sum_{i=1}^{X_{\mathrm{B}}}\left\{c_{c,i,t-\Delta t}\left[1 - \exp\left(-\frac{\Delta t S'_{c,i,t-\Delta t}k_{c,i,t-\Delta t}}{H'_{c,i,t-\Delta t}}\right)\right]\right\}}{1 + \sum_{i=1}^{X_{\mathrm{B}}}\left\{\frac{H'_{c,i,t-\Delta t}}{S'_{c,i,t-\Delta t}}\left[1 - \exp\left(-\frac{\Delta t S'_{c,i,t-\Delta t}k_{c,i,t-\Delta t}}{H'_{c,i,t-\Delta t}}\right)\right]\right\}}. \tag{6}$$

Here $H'_{c,i}$ is the dimensionless Henry's constant for chemical compound $c$ in size bin $i$

$$H'_{c,i} = m_v c_{w,i} R * T H_c , \tag{7}$$

where $m_v$ $(\mathrm{mol\,m}^{-3})$ is the molecular weight of water, $c_{w,i}$ $(\mathrm{mol\,m}^{-3})$ is the mole concentration of liquid water in aerosol size bin $i$, $R* = 8.206\,\mathrm{m}^3\,\mathrm{atm}\,\mathrm{K}^{-1}\,\mathrm{mol}^{-1}$ is the universal gas constant, $T$ (K) is the ambient temperature, and $H_c$ $(\mathrm{mol\,kg}^{-1}\,\mathrm{atm}^{-1})$ is the Henry's law constant estimated by the thermodynamic model PD-FiTE (Topping et al., 2009). Finally, the new particle mole concentration $c_{c,i,t}$ is given by

$$c_{c,i,t} = \frac{H'_{c,i,t-\Delta t}C_{c,t}}{S'_{c,i,t-\Delta t}} + \left(c_{c,i,t} - \frac{H'_{c,i,t-\Delta t}C_{c,t}}{S'_{c,i,t-\Delta t}}\right)\exp\left(-\frac{\Delta t S'_{c,i,t-\Delta t}k_{c,i,t-\Delta t}}{H'_{c,i,t-\Delta t}}\right), \tag{8}$$

which is then translated to number and mass concentrations. Evaporation of gases from aerosol particle surfaces, with water being an exception, is not considered.

### 2.2.3   Dry deposition and gravitational settling

Dry deposition removes aerosol particles from air when they collide with a surface and stick to it. Here the original scheme in SALSA allowing dry deposition on horizontal surfaces was extended by including deposition also on vertical solid surfaces (e.g.

building walls) and resolved-scale vegetation. Deposition on sub-grid vegetation (e.g. grass surface) is not yet implemented. By





default, dry deposition velocity $v_d$ $(\mathrm{m\,s^{-1}})$ is calculated applying the size-segregated scheme by Zhang et al. (2001) (hereafter Z01), which is the most applied dry deposition scheme in numerical studies. For size bin $i$:

$$v_{d,i} = \underbrace{\frac{(\rho_p - \rho_a)\overline{D}_i^2\, g\, G_i}{18\eta_a}}_{\text{settling velocity, } v_{c,i}} + \epsilon_0 u_* \exp(-St_i^{1/2}) \left[ \underbrace{Sc_i^{-\gamma}}_{\text{Brownian diffusion}} + \underbrace{\left(\frac{St_i}{\alpha + St_i}\right)^{\beta}}_{\text{impaction}} + \underbrace{\frac{1}{2}\left(\frac{\overline{D}_i}{A}\right)^2}_{\text{interception}} \right], \tag{9}$$

where $\rho_p$ and $\rho_a$ are the particle and air densities $(\mathrm{kg\,m^{-3}})$, $g$ $(\mathrm{m\,s^{-2}})$ is the gravitational acceleration, $G_i$ is the Cunningham

slip correction factor, $\eta_a$ $(\mathrm{kg\,m^{-1}\,s^{-1}})$ is the dynamic viscosity of air, $\epsilon_0 = 3$ and $\beta = 2$ are empirical constants, $u_*$ $(\mathrm{m\,s^{-1}})$ is the friction velocity of above a surface, $St_i$ is the Stokes number, $Sc_i$ is the particle Schmidt number, $\gamma$ and $\alpha$ are empirical constants that depend on the surface type, and $A$ is the characteristic radius of the different surface types and seasonal categories. Note that the aerodynamic resistance in the original Z01 formulation is not considered here as LES resolves the aerodynamic effect explicitly. For solid surfaces, $u_*$ is solved within PALM applying a stability-adjusted logarithmic wind

profile, whereas for the resolved-scale vegetation an estimation $u_* = \sqrt{C_D}\, U$ (Prandtl, 1925), where $C_D$ is the canopy drag coefficient and $U = \sqrt{u^2 + v^2 + w^2}$ is the three-dimensional wind speed, is applied. Z01 has been suggested to overestimate $v_d$ for submicron particles (Petroff and Zhang, 2010; Mingxuan et al., 2018), and therefore as an alternative to Z01, the formulation by Petroff and Zhang (2010) (hereafter P10) for the deposition velocity can be used (see Supplement information (SI), Sect. S1). The different parametrisations Z01 and P10 for $v_d$ over built surfaces and deciduous broadleaf trees during leaf-on

period are visualised in Fig. 1.

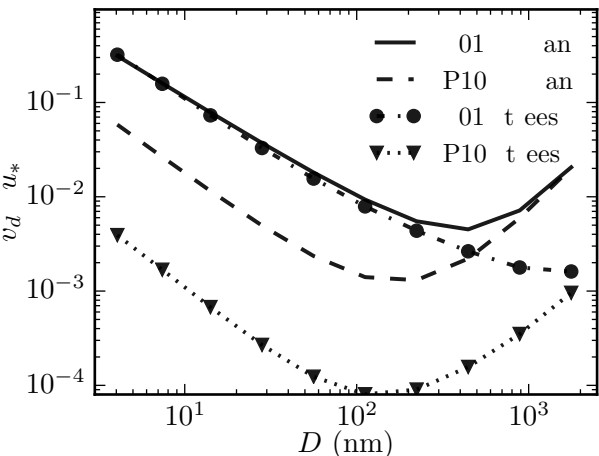

**Figure 1.** Normalised deposition velocity $v_d$ / $u_*$ as a function of aerosol particle diameter $D$ (nm) for urban surfaces (solid and dashed lines) and deciduous broadleaf trees (dash-dot line with circles and dotted line with triangles) using the parametrisation by Zhang et al. (Z01, 2001) and Petroff and Zhang (P10, 2010).

(c) Author(s) 2018. CC BY 4.0 License.





Dry deposition on vegetation creates a local sink term

$$\frac{\partial n_i}{\partial t} = -\text{LAD}\, v_{d,i} n_{i,t-\Delta t}\,, \tag{10}$$

whereas dry deposition on horizontal surfaces and building walls is implemented by means of surfaces fluxes

$$F_{n_i} = -v_{d,i} n_{i,t-\Delta t}\,. \tag{11}$$

The same equations apply for $m_{c,i}$. When not in contact with a surface, only gravitational settling contributes to dry deposition and generates a downward flux of particles, which is however mainly important for large particles ($D > 1.0\,\mu\text{m}$) (Zhang et al., 2001; Petroff and Zhang, 2010). Dry deposition and gravitational settling are currently calculated only for aerosol particles, and not for gaseous components.

### 2.2.4   New particle formation

In the model evaluation represented here, nucleation is assumed to have already occurred (Rönkkö et al., 2007; Uhrner et al., 2007) and the nucleation mode aerosol particles are given to the model as an input. Notwithstanding, new particle formation by sulphuric acid can be taken into account by calculating the apparent rate of formation of 3 nm sized aerosol particles according to the parametrisation by Kerminen and Kulmala (2002), Lehtinen et al. (2007) or Anttila et al. (2010). To calculate the "real" nucleation rate, user can choose between the binary (Vehkamäki et al., 2002), ternary (Napari et al., 2002a, b), kinetic (Sihto

et al., 2006; Riipinen et al., 2007) or activation-type (Riipinen et al., 2007) nucleation.

### 2.2.5   Emissions

Aerosol particle emissions can be given to the model as an input applying three levels of detail (LOD): parametrised (LOD1, units $\text{kg}\,\text{m}^{-2}\,\text{s}^{-1}$) or detailed (LOD2, units $\text{m}^{-2}\,\text{s}^{-1}$) 2-dimensional surface fluxes, or 3-dimensional sources (LOD3, units $\text{m}^{-3}\,\text{s}^{-1}$). Using LOD1, aerosol emissions are given as particulate mass (PM) emissions, from which the size-segregated

number emissions $E_{n_i}$ are calculated within the model implementing default aerosol size distributions and mass compositions for each emission category EC (e.g. traffic, domestic heating, etc.). LOD2 and LOD3 emission data include $E_{n_i}$ and the mass composition per each EC, based on which the mass emission per size bin $i$ and chemical component $c$ are then calculated within the model. Gaseous emissions can be specified using any LOD. The time dependency of the aerosol emissions has not been implemented yet.

## 2.3   Model coupling and steering

SALSA is integrated to PALM as an optional PALM-4U module, which directly utilises the momentum and scalar concentration fields of the parent model as input. The aerosol source/sink terms are resolved sequentially at a user-specified frequency $f_{\text{SALSA}}$, while the prognostic equations and thus transport of aerosol number and mass as well as gas concentrations are resolved at every LES time step $\Delta t_{\text{LES}}$ in PALM. Molecular diffusion is assumed negligible compared with the turbulent diffusion and

is thus ignored.

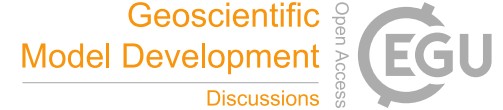



Since water is a default chemical component in SALSA, PALM needs to be run in the humid mode (i.e. calculate the prognostic equation for specific humidity $q$). The particle water content $m_{H_2O,i}$ per size bin $i$ can be represented either as a prognostic variable, or as a diagnostic variable and calculated at each $\Delta t_{SALSA}$ based on the equilibrium solution using the ZSR method (Stokes and Robinson, 1966). The feedback on the temperature and humidity due to condensation of water vapour on particles can be switched off. Moreover, SALSA can be run together with the available PALM-4U chemistry module to transfer the gas concentrations, while the impact of aerosol particles on radiative transfer has not been implemented yet.

## 2.4 Computational expenses

**Table 2.** The relative change in the total computational time over a $20\,\mathrm{m} \times 20\,\mathrm{m} \times 20\,\mathrm{m}$ modelling domain with different configurations for SALSA. The number of simulated size bins $X_B = 10$, time step of the LES model $\Delta t \approx 2\,\mathrm{s}$ and the total simulation time $1000\,\mathrm{s}$. $X_{CC}$ stands for the number of chemical components and $\Delta X_{PV}$ for the change in the change in the number of prognostic variables.

| Run | $X_{CC}$ | $\Delta X_{PV}$ | Aerosol processes | $H_2O$ advection | $\Delta t_{SALSA}$ | Change in the computational time (%) |
|---|---|---|---|---|---|---|
| 1 | $H_2SO_4$ | 35 | - | yes | $\Delta t$ | + 390 |
| 2 | $H_2SO_4$ | 25 | - | no, ZSR method | $\Delta t$ | + 530 |
| 3 | $H_2SO_4$ | 35 | coagulation | yes | $\Delta t$ | + 780 |
| 4 | $H_2SO_4$ | 35 | nucleation | yes | $\Delta t$ | + 430 |
| 5 | $H_2SO_4$ | 35 | dry deposition (Z01) | yes | $\Delta t$ | + 410 |
| 6 | $H_2SO_4$ | 35 | dry deposition (P10) | yes | $\Delta t$ | + 410 |
| 7 | $H_2SO_4$ | 35 | condensation | yes | $\Delta t$ | + 400 |
| 8 | $H_2SO_4$, OC | 45 | condensation | yes | $\Delta t$ | + 510 |
| 9 | $H_2SO_4$, OC, $HNO_3$ | 55 | condensation | yes | $\Delta t$ | + 600 |
| 10 | $H_2SO_4$, OC, $HNO_3$, $NH_3$ | 65 | condensation | yes | $\Delta t$ | + 820 |
| 11 | $H_2SO_4$, OC, $HNO_3$, $NH_3$, BC | 75 | all | yes | $\Delta t$ | + 1370 |
| 12 | $H_2SO_4$, OC, $HNO_3$, $NH_3$, BC | 75 | all | yes | $2\Delta t$ | + 1130 |
| 13 | $H_2SO_4$, OC, $HNO_3$, $NH_3$, BC | 75 | all | yes | $10\Delta t$ | + 810 |

Each $n_i$, $m_{c,i}$ and gaseous compound introduces a new prognostic variable that is transported by the flow in PALM. Increasing the number of prognostic variables $X_{PV}$ from the default value of $X_{PV} = 6$ (wind components $u$, $v$, $w$ and scalars $e$, $\theta$ and $q$) to

$$X_{PV} = 6 + \Delta X_{PV} = 6 + X_B(X_{CC} + 1) + X_G \,, \tag{12}$$

where $X_B$ is the number of size bins, $X_{CC}$ the total number of chemical components (aerosol-phase) and $X_G = 5$ the total number of gaseous compounds, increases the computational load tremendously. To estimate the increase in computational



costs caused by significantly increasing $X_{PV}$, simulations over a simple test domain of 20 m × 20 m × 20 m (See SI, Fig. S1) were conducted with varying set-ups for SALSA. The relative changes in computational load per simulation are given in Table 2. Adding $X_B = 10$ size bins composed of $X_{CC} = 2$ chemical components (water always present) introduces $\Delta X_{PV} = 35$ new prognostic variables and increases the original computational time by nearly a factor of four (run 1). Calculating the aerosol

water content at each $\Delta t_{SALSA}$ instead of treating it as prognostic variables is even more demanding (run 2). Of all aerosol dynamic processes, coagulation is the most expensive (run 3). Including more chemical components further increases the computational time (run 8–13), which can be however notably decreased by lengthening $\Delta t_{SALSA}$ (run 12–13). Considering the longer time scales of aerosol dynamic processes compared to dispersion, $\Delta t_{SALSA} = 10\Delta t$ is considered to be reasonable in urban simulations with a grid resolution of ∼1 m and $\Delta t \sim 0.1$.

## 2.5    Initialisation of the aerosol number and mass size distribution

The initial aerosol size distribution is defined by setting the number concentration of particles in each bin $n_i$ of which the volume $v_{c,i}$ and mass concentrations $m_{c,i}$ are calculated based on the geometric mean diameter $\overline{D}_i$. Aerosol emissions are defined similarly. In other words, the total number concentration is preserved in the initialisation whereas uncertainties arise when estimating $m_{c,i}$ or $v_{c,i}$.

Limiting $X_B$ in a sectional aerosol module is a simple method to reduce computational costs and memory demand. However, this results in an inevitable loss in accuracy as the aerosol size range covers many orders of magnitude from few nanometres to several micrometres. To test the sensitivity of the representation of the aerosol number and mass size distribution to $X_B$, four different configurations are tested (Fig. 2). All configurations cover particles from 3 nm to 2.5 μm and the subrange 1 includes particles up to 10 nm. The default configurations contains $X_B = 10$ with two bins in the subrange 1. The second

configuration contains $X_B = 8$ and only one bin in the subrange 1, whereas the third configuration contains two additional bins in the subrange 2 when compared to the default configuration. Additionally, an ideal configuration with $X_B = 50$ was tested.

The total aerosol particle volume concentration $V$ is highly sensitive to $X_B$ and the rate of overestimation increases with a decreasing $X_B$ (Fig. 2). Overestimating particle volume causes errors in, for instance, calculating the coagulation kernel, gas-to-particle mass transfer and deposition velocity. Furthermore, the ability of a sectional module to capture narrow features

in a size distribution (e.g. in Fig. 2c) improves with higher $X_B$. To compromise between computational costs and modelling accuracy, $X_B = 10$ is used in this evaluation study.

## 3    Model evaluation set-up

### 3.1    Case description

Performance of SALSA module in PALM is evaluated against measurements on the vertical variation of the aerosol number size

distribution and concentrations in a street-canyon (Pembroke Street) in central Cambridge, United Kingdom, over consecutive 24 hours on March 20–21, 2007 (Kumar et al., 2008, 2009). During the measurement campaign, the predominant wind direction



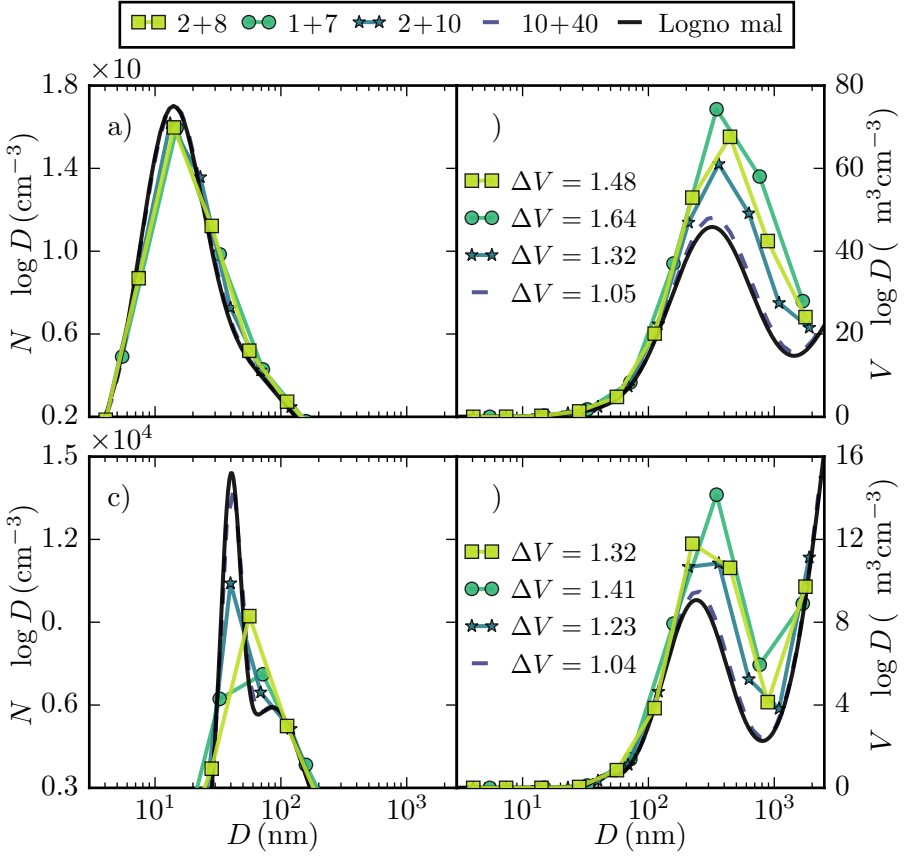

**Figure 2.** A sectional representation of the aerosol number $dN/d\log D$ $(\mathrm{cm}^{-3})$ (a and c) and volume $dV/d\log D$ $(\mu\mathrm{m}^3\,\mathrm{cm}^{-3})$ (b and d) size distribution as a function of particle diameter $D$ (nm) in SALSA for typical polluted urban (a–b) and hazy rural conditions (c–d) (Zhang et al., 1999). Top legend: [number of size bins in the subrange 1] + [number of size bins in the subrange 2]. The continuous log-normal size distribution is given with a solid black line. $\Delta V$ is the total volume concentration relative to the continuous log-normal size distribution.

was from northwest and perpendicular to the street canyon. Furthermore, there is a large pedestrian area upwind of the site with no traffic emissions, and hence emissions from adjacent streets unlikely affected the measurements. The building height is around 14–18 m on the upwind and 11–15 m on the downwind side of the street canyon (Fig. 3).

Aerosol size distributions in the size range $D = 5-2738$ nm were measured pseudo-simultaneously at four heights ($z = 1.00$, 2.25, 4.62 and 7.37 m above ground level (AGL)) on the northwestern side of Pembroke street using a fast-response differential mobility spectrometer (DMS500). Traffic volumes along the street were simultaneously measured. Moreover, 30-minute-averaged meteorological data, including wind speed ($U$) and direction, ambient air temperature ($T$) and relative humidity (RH), were measured 40 m AGL at some 500 m from the sampling site. For more information on the measurements, refer to Kumar et al. (2008).



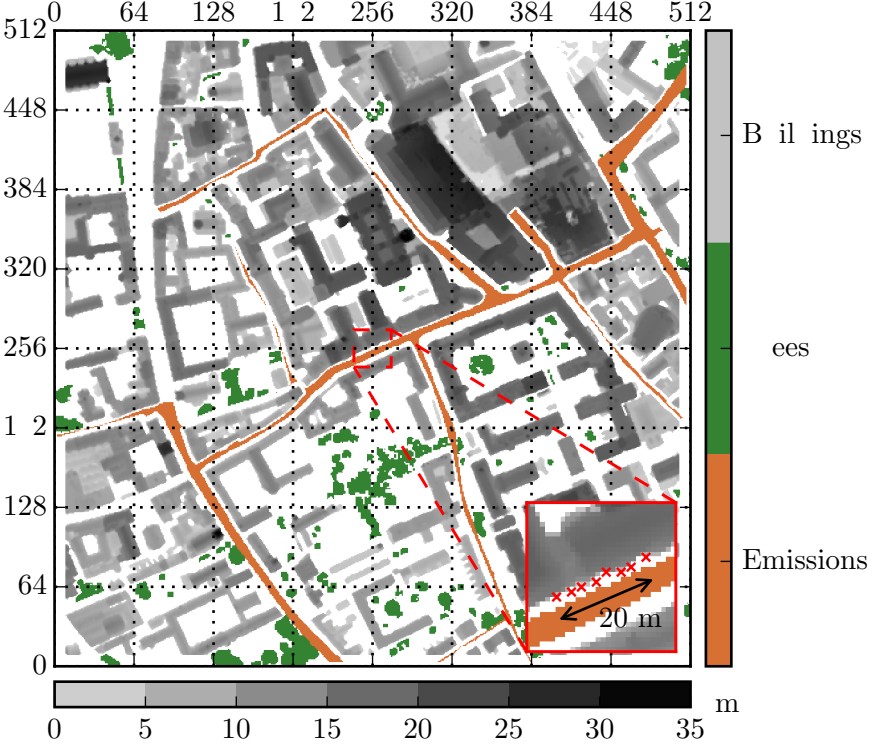

**Figure 3.** Visualisation of the simulation domain. The building height (m) is shown in colors of grey, and the location of trees and emissions in green and copper, respectively. The evaluation domain is marked with a red square. Red crosses in the zoomed figure indicate the horizontal points at which the model output is evaluated against measurements. The grid shows the horizontal model grid. Data sources: elevation maps – Environment agency (UK) data archive; land use footprints – Ordnance Survey 2014.

The evaluation is done for three different periods: 8.30–9.30 am (morning), 9.00–10.00 pm (evening) and 3.00–4.00 am (night-time). No daytime evaluation is presented here in order to minimize the role of thermal and vehicle-induced (VIT) turbulence on pollutant transport. The evening and night-time periods represent time after sunset while the morning measurements were conducted under partly cloudy conditions.

## 3.2 Model domain and morphological data

Simulations are conducted over a domain of $512 \times 512 \times 128$ grid box with the measurement site approximately at the centre of the domain (Fig. 3). A uniform grid spacing of $\Delta_{x,y,z} = 1.0$ m is applied within the lowest 96 m, and above the vertical grid $\Delta_z$ is stretched by a factor of 1.04, resulting in a total domain height of around 164 m and a maximum $\Delta_{z,\mathrm{max}} \approx 3.5$ m.

The building-height and vegetation maps for the study area were constructed from 1-m horizontal resolution digital surface (DSM) and terrain (DTM) models (Environment agency UK data archive) following Kent et al. (2018). First, the DTM was subtracted from the DSM to set the terrain height to zero. Next, buildings were separated from other surface elements using





a building footprint dataset of the OS MasterMap® Topography Layer (Ordnance Survey 2014). Vegetation map was formed from the remaining pixels by first removing the residue pixels around buildings and then performing dilation of the raster map to remove holes and unify vegetated areas. Only vegetation elements higher than $z_{v,\,min} = 4.0\,\mathrm{m}$ were included in the simulations. They were modelled as springtime, deciduous broadleaf trees, with a constant leaf area density $\mathrm{LAD} = 0.6\,\mathrm{m}^2\,\mathrm{m}^{-3}$ from $z_{v,\,min}$

to the tree top. Excluding the details of local vegetation is acceptable since there are no trees close to the measurement site and overall the amount of vegetation is low.

Only road traffic lanes are defined as source areas for aerosol particles and gaseous compounds. The emission map (Fig. 3) was created by first extracting the roads, tracks and paths from the OS MasterMap® Topography Layer, and then manually removing pedestrian areas and small streets. Finally, raster erosion was applied to the remaining map to result in a lane width

of 6–7 m on Pembroke street.

### 3.3   Pollutant boundary conditions: emissions and background concentrations

**Table 3.** Emission factors (EF) applied in the simulations for all gaseous compounds and aerosol number $n$.

|     | $H_2SO_4$ | $HNO_3$ | $NH_3$ | NVOC | SVOC | $n$ |
|-----|-----------|---------|--------|------|------|-----|
|     |           | ($\mathrm{g\,km^{-1}\,vehicle^{-1}}$) | | | | ($\mathrm{km^{-1}\,vehicle^{-1}}$) |
| EF  | $2.5\times10^{-4}$ | 0.0 | $4.2\times10^{-2}$ | 0.0 | $2.5\times10^{-3}$ | $1.33\times10^{14}$ |

In the simulations, a total aerosol number emission factor $\mathrm{EF}_n = 1.33 \times 10^{14}\,\mathrm{km}^{-1}\,\mathrm{vehicle}^{-1}$ is used (Table 3), which is an estimate specific to the measurement site (Kumar et al., 2009). $\mathrm{EF}_n$ was distributed to a representative aerosol number size distribution with the shape estimated from the measured size distribution at the lowest level $z = 1.0$ m during each simulation

time (see SI, Sect. S3). Aerosol emissions are assumed to be composed of mainly black (48%) and organic carbon (48%), and some $H_2SO_4$ (4% of the total mass) (Maricq, 2007; Dallmann et al., 2014). Emission factors of gaseous compounds, instead, are calculated using the fleet weighted road transport emission factors for 2008 by the National Atmospheric Emissions Inventory (NAEI, Walker, 2011) and the following fleet composition: 75 % petrol and 19 % diesel passenger cars, 1 % buses, 3 % light and 1 % heavy duty diesel vehicles, and 1 % motorcycles. Since no $\mathrm{EF}_{H_2SO_4}$ or $\mathrm{EF}_{SVOC}$ is given by NAEI, the following estimates

were applied: $\mathrm{EF}_{H_2SO_4} = 0.1\,\mathrm{EF}_{SO_2}$ (Arnold et al., 2006, 2012; Miyakawa et al., 2007) and $\mathrm{EF}_{SVOC} = 0.01\,\mathrm{EF}_{NMOG}$ (Zhao et al., 2017), where NMOG stands for non-methane organic gases. The latter is a rather conservative compared to emission rates applied by Albriet et al. (2010) for a light duty diesel truck. Both aerosol and gaseous emissions are introduced as constant fluxes per unit area.

The background aerosol particle number and trace gas concentrations are produced with the trajectory model for Aerosol

Dynamics, gas and particle phase CHEMistry and radiative transfer (ADCHEM, Roldin et al., 2011). Similar to Öström et al. (2017), ADCHEM was operated as a one-dimensional column trajectory model along HYSPLIT (Stein et al., 2015) air mass trajectories. In total, the gas- and aerosol particle composition were simulated along 48 trajectories arriving to central Cam-



bridge between March 20 at 00:00 and March 21 at 23:00 (one every hour). All air mass trajectories started 5-days upwind Cambridge over the Arctic ocean (see SI, Fig. S5). The anthropogenic trace gas emissions along the trajectories were taken from the European Monitoring and Evaluation Programme (EMEP) emission inventory for 2007 and the size resolved primary particle emissions from the global emission inventory from Paasonen et al. (2016). These vertical profiles of the background

concentrations (SI, Sect. S5) are introduced to the simulation domain by a decycling method, in which the constant background concentrations are fixed at the lateral boundaries.

### 3.4 Flow boundary conditions

**Table 4.** Prevailing wind speed $U$, air temperature $T$ and relative humidity RH at $z = 40$ m AGL, the applied external pressure gradient force and traffic rates for each simulation hour. Wind direction is always from northeast.

| Simulation | $U$ (m s$^{-1}$) | $T$ (K) | RH (%) | Pressure gradient in $x, y$-directions (Pa m$^{-1}$) | Traffic rate (veh hour$^{-1}$) |
|---|---|---|---|---|---|
| Morning | 4.30 | 277 | 64 | -0.00630, 0.00630 | 895 |
| Evening | 3.94 | 274 | 90 | -0.00515, 0.00515 | 380 |
| Night | 2.24 | 272 | 93 | -0.00164, 0.00164 | 306 |

In all simulations, a neutral atmospheric stratification is assumed for simplicity as no information on the atmospheric stratification nor boundary layer height was available. Thus, a constant $\theta = T(z = 40\,\text{m})$ (Table 4) is applied throughout the domain.

The flow is driven by an external pressure gradient force above $z = 120$ m. The gradient was set so that the horizontal mean $U\,(z = 40\,\text{m})$ over the whole simulation domain equals ($\pm 0.1\,\text{m s}^{-1}$) the measured $U$ (Table 4). Furthermore, a domain height of 164 m was set for all simulations. This is $> 13h$, where $h = 12.08$ m is the mean building height over the domain, which should be enough to correctly resolve the small-scale turbulent structures within the urban canopy (Coceal et al., 2006).

Cyclic lateral boundary conditions are applied for the flow, $q$ and $e$, which is reasonable since the surroundings do not notably

differ from the simulation domain. A Neumann (free-slip) boundary condition is applied at the top boundary and also at the bottom and top for all scalars. The roughness height is $z_0 = 0.05$ m and the drag coefficient applied for the trees $C_D = 0.5$.

### 3.5 Simulations

Baseline simulations used to evaluate the performance of the model in the morning, evening and at night are conducted with the default number of aerosol size bins $X_B = 2 + 8$ (see Sect. 2.5). All aerosol processes, except nucleation, are switched on,

and the following chemical components are included: $H_2SO_4$, OC, BC, $HNO_3$ and $NH_3$. All aerosol particle are assumed to be internally mixed and hygroscopic, and thereby no subrange 2b was applied.

In addition to the base run, the sensitivity to different aerosol processes and the number of size bins $X_B$ was examined for the morning simulation. Firstly, the following four simulations with $X_B = 2 + 8$ are conducted: no aerosol processes (NOAP),



only coagulation (COAG), only dry deposition (scheme Z01) on solid surfaces and vegetation (DEPO), and only condensation (COND). In the first three, particles are assumed to constitute only of OC in order to limit computational costs, given that coagulation and dry deposition do not depend on the aerosol composition. COND, instead, is performed with an identical set-up as the baseline simulation, except that other processes were switched off. Secondly, the sensitivity to $X_B$ is tested by

replicating the baseline morning simulation with less $X_B = 1 + 7$ (LB) and more bins $X_B = 2 + 10$ (MB).

The advection of both momentum variables and scalars was based on the 5th-order advection scheme by Wicker and Ska-marock (2002) together with a third-order Runge-Kutta time stepping scheme (Williamson, 1980). The pressure term in the prognostic equations for momentum was calculated using the iterative multigrid scheme (Hackbusch, 1985). In order to en-able similar flow conditions for all simulations, feedback to PALM was switched off, i.e. changes in specific humidity due to

condensation of water on aerosol particles were not allowed. Therefore $q$ also remained constant. Here $\Delta t_{\mathrm{SALSA}} = 1.0$ s in all simulations, which is a safe choice since the turbulence time scale is smaller than any aerosol process time scale (Kumar et al., 2008).

Simulations were conducted with the PALM model revision 3125. This was a model version prior to the 6.0 release, but reproducibility with version 6.0 was ensured by repeating the NOAP simulation. All simulations were first run for two hours

to create a quasi-stationary state of the flow, after which SALSA was switched on and run for 70 minutes. Data output was collected within the last 60 minutes with a 0.5–1 Hz frequency. Simulations were performed on the Centre for Scientific Computing (CSC) Taito supercluster. Using $64 \times 64$ Intel Haswell processor cores, one 70-minutes-long simulation with SALSA required between 17 h (NOAP) and 52 h (MB) computing time.

## 4   Results

Modelled aerosol number concentrations were compared against measurements at eight horizontal points on the northern side of the street canyon within the evaluation domain of 30 m × 30 m (Fig. 3). All eight profiles were analysed to include the possible error in defining the measurement location and also to illustrate the variation in concentrations at different adjacent points in a street canyon. All modelled and measured values are hourly averaged. Statistical analysis of the model is not conducted as it would not be statistically representative with the small amount of data points available for evaluation.

### 25   4.1   Baseline simulations

To give a general picture of aerosol particle concentrations and dispersion in this study, Fig. 4 illustrates the modelled total aerosol number concentrations $N_{\mathrm{tot}}$ and wind speed $U$ at $z = 3.5$ m AGL for all baseline simulations. The horizontal distribu-tion of $N_{\mathrm{tot}}$ is shown to follow that of emissions (see Fig. 3) and, for instance, courtyards remain relatively clean. Nevertheless, wind controls the dispersion, which is seen as up to 70 % higher $N_{\mathrm{tot}}$ inside the street canyons for the calmer night-time com-

pared to the more windy evening simulation (see Fig. S7 in SI) despite the lower emission rates at night. Interestingly, pollutant accumulation occurs close by the measurement site within the evaluation domain.



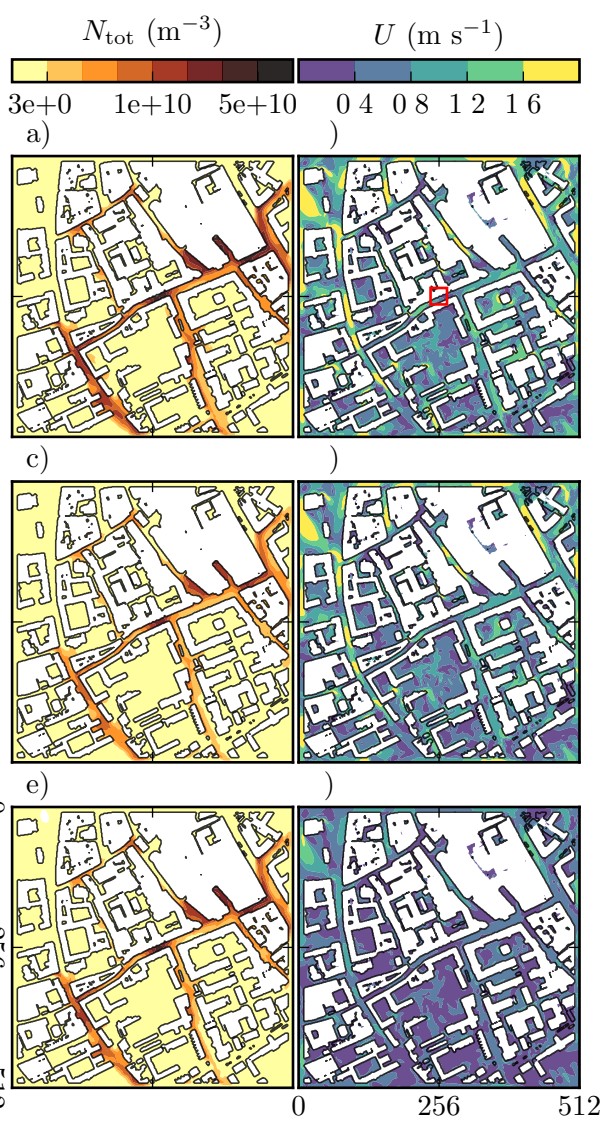

**Figure 4.** Total aerosol number concentration $N_{\text{tot}}$ ($\text{m}^{-3}$, left column) and wind speed $U$ ($\text{m s}^{-1}$, right column) at $z = 3.5$ m for the morning (a, b), evening (c, d) and night-time simulation (e, f) over the whole simulation domain of 512 m × 512 m. Evaluation domain (see Fig. 3) is marked with a red square in b).

The modelled mean vertical profiles of $N_{\text{tot}}$ ($\text{m}^{-3}$) at all eight horizontal points compare well against the measured values (Fig. 5). Despite the slight overestimation of $N_{\text{tot}}$ in the evening (Fig. 5b), concentrations are in the same order of magnitude. Furthermore, the variation between the mean values of all eight modelled profiles is larger than the variation between their





mean value and measured $N_{\text{tot}}$. The rate of change of $N_{\text{tot}}$ in vertical is correctly modelled except for a measured increase in concentrations within the lowest 2 m.

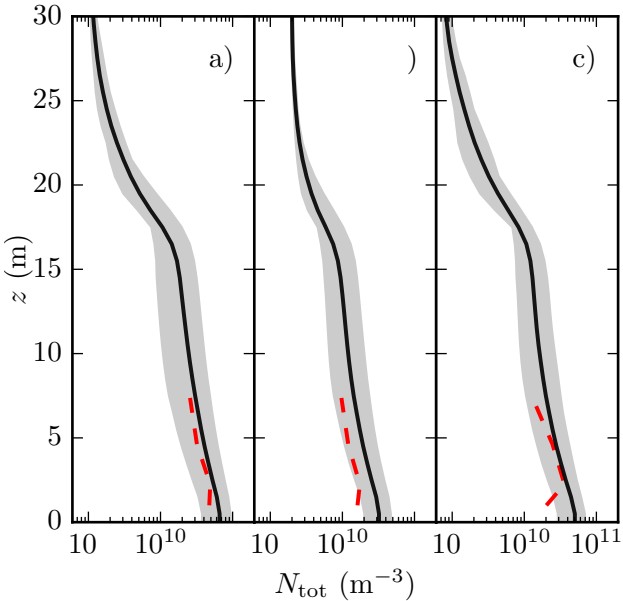

**Figure 5.** Measured (red dashed line) and modelled (black solid line and grey shaded area) vertical profiles of total aerosol number concentration $N_{\text{tot}}$ (m$^{-3}$) for the a) morning, b) evening and c) night-time simulation. The grey shaded area shows the range of the mean vertical profiles 1–8 within the evaluation domain and the solid line their mean value.

The coarse sectional representation with $X_{\text{B}} = 10$ means some details, such as the drop in concentrations at $D \approx 60$ nm, cannot be always captured by the model (Fig.6). Furthermore, the number of particles larger than 20 nm is underestimated in the night-time simulation at $z = 2.5$ m and $z = 4.5$ m (Fig. 6 b and c), which could stem from omitting some emissions elevated from the surface, such as tail-pipes of trucks. Overall, the modelled size distributions display very similar shapes to the emission size distributions, showing that the result is very sensitive to the quality of the input emission data.

Moreover, as the LES technique lacks reliability close to the walls, a mismatch with the measurements near the surface is to be expected. Maronga et al. (2015), for instance, showed that the turbulent flow over a homogeneous surface is not well-resolved for the lowest six grid points, which corresponds to the lowest 5 m in these simulations. In that context, the modelled concentration fields agree exceptionally well with the measurements.





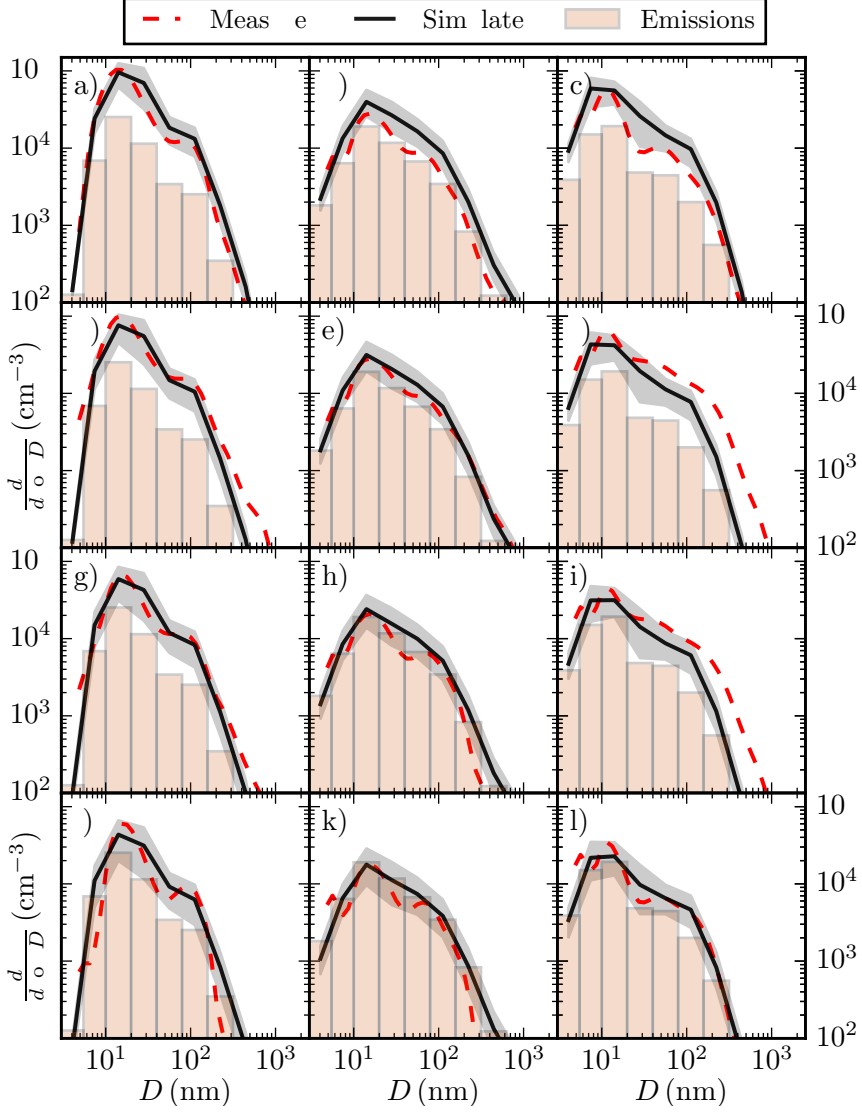

**Figure 6.** Measured (red dashed line) and simulated (black lines) aerosol number size distribution $dN/d\log D$ (cm$^{-3}$) as a function of particle diameter $D$ (nm) in the morning (first column: a, d, g, j), evening (second column: b, e, h, k) and at night (third column: c, f, i, l) at levels $z = 0.5, 2.5, 4.5$ and $7.5$ m (top to bottom). The shape of the number size distribution for the emissions is given with bars (not in units cm$^{-3}$). The grey shaded area shows the range of the horizontal points 1–8 within the evaluation domain and the line their mean value.



## 4.2 Sensitivity tests

### 4.2.1 Role of different aerosol processes

In the temporal and spatial scales applied in the simulations, dry deposition changes the total aerosol number concentrations most, with a relative difference $\Delta N_{\text{tot}} < -20$ % especially in areas with vegetation but also in the wake of buildings (Fig. 7).

5 Coagulation (COAG) changes $N_{\text{tot}}$ only by less than 1 %. The impact of condensation and dissolutional growth (COND) on $N_{\text{tot}}$ is negligible, as expected, since condensation only grows particles (Kumar et al., 2011).

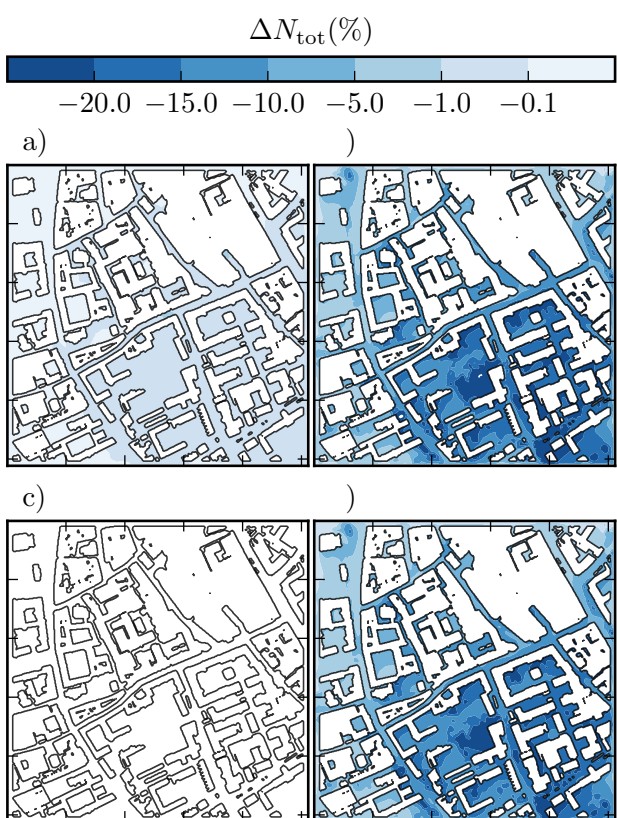

**Figure 7.** Relative difference in the total aerosol number concentration $\Delta N_{\text{tot}}$ (%) at $z = 3.5$ m compared to NOAP for the a) COAG, b) DEPO, c) COND and d) baseline simulation in the morning.

Neglecting all aerosol processes overestimates $N_{\text{tot}}$ (Fig. 7a), and therefore including dry deposition in the model is essential for realistic $N_{\text{tot}}$ shown as the agreement with measurements (Fig. 7d and e). Above the roof level ($z \gtrsim 15$ m), the role of dry deposition starts to weaken, which is also attributable to lower aerosol concentrations. Smallest aerosol particles are most

10 strongly affected by the aerosol processes independently from the modelling height (Fig. 9): This is because more efficient Brownian diffusion leads to higher deposition velocities $v_d$ (see Fig. 1) and coagulation rates. Furthermore, smallest particles





grow by condensation and dissolutional growth, which instead leads to less efficient removal by dry deposition. The impact of dry deposition and, to less extent, coagulation, decrease with height, and above the roof level the observed $\Delta N_{\text{tot}}$ is likely due aerosol processes acting upwind of the measurement site.

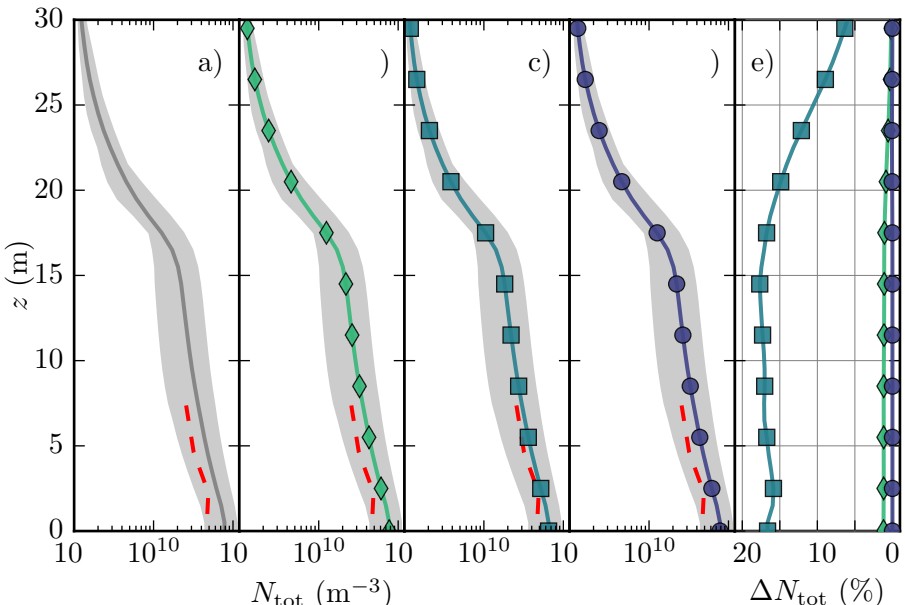

**Figure 8.** Measured (red dashed line) and simulated vertical profiles of the total aerosol number concentration $N_{\text{tot}}$ ($\text{m}^{-3}$) for the a) NOAP (grey solid line), b) COAG (diamonds) and c) DEPO (squares) and d) COND (circles) simulation in the morning. Relative difference $\Delta N_{\text{tot}}$ (%) of b)–d) to a) is shown in e). The grey shaded area shows the range of the mean vertical profiles 1–8 within the evaluation domain and the line their mean value.

While condensation and dissolutional growth do not directly affect the number concentrations, the total mass and chemical composition of aerosol particles are shown to change. Over the whole evaluation domain, condensation and dissolutional growth increase $\text{PM}_{\text{tot}}$ by over 10 % below the roof height (Fig. 10). Comparing the initial chemical composition of the background aerosol concentrations and emissions (Table 5) with the modelled composition shows that especially the mass fraction of nitrates has increased, from 0 to 8 %. This increased particulate mass of nitrates originates solely from condensation of background gaseous $\text{HNO}_3$ as there are no traffic related emissions of gaseous $\text{HNO}_3$. The simulated mass fraction of BC is very close to that of the aerosol emissions, while other mass fractions that also change due to condensation and dissolutional growth vary more. Deposition decreases $\text{PM}_{\text{tot}}$ but the relative change is clearly lower than for $N_{\text{tot}}$ as smallest particles, which are most affected by dry deposition, represent only a tiny share of the total mass.





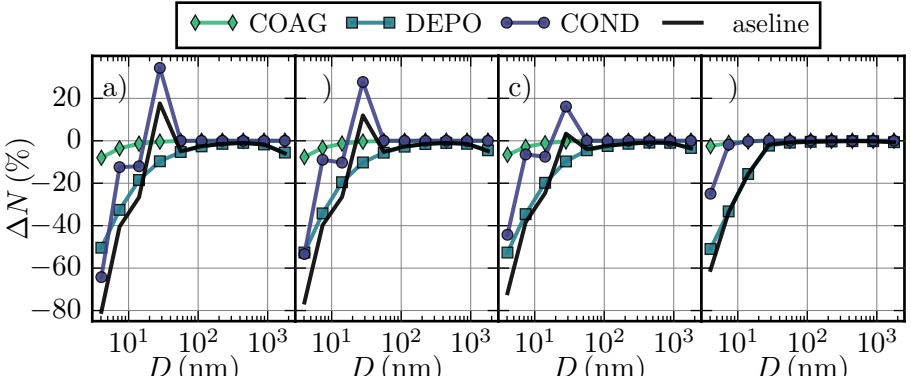

**Figure 9.** Relative difference in the aerosol number concentration $\Delta N$ (%) compared to NOAP as a function of aerosol particle diameter $D$ (nm) at levels a) $z = 3.5$ m, b) $z = 10.5$ m, c) $z = 20.5$ m and d) $z = 40.5$ m in the morning.

**Table 5.** Mass fractions of different chemical compounds for the aerosol background, emissions and simulated concentrations for the COND simulation. The values are averaged over the whole evaluation domain within $z < 30$ m.

|  | $SO_4^{2-}$ | OC | BC | $NO_3^-$ | $NH_4^+$ |
|---|---|---|---|---|---|
| Background | 0.09 | 0.24 | 0.64 | 0.0 | 0.03 |
| Emission | 0.04 | 0.48 | 0.48 | 0.0 | 0.0 |
| Simulated: COND | 0.05 | 0.36 | 0.49 | 0.08 | 0.01 |

### 4.2.2 Number of size bins

Further decreasing the number of aerosol size bins $X_B$ is a tempting method in order to reduce the computational load. Indeed, the total CPU time is reduced by -24 % when $X_B = 1 + 7$ (LB), while setting $X_B = 2 + 10$ (MB) increases the CPU time by +18 % compared to the baseline simulation in the morning. However, as shown in Sect. 2.5 and Fig. S8 (see SI), the capability

5   to describe the details in the aerosol size distribution drops rapidly when decreasing $X_B$.

Despite the background $N_{tot}$ and total aerosol number emissions $EF_n$ being equal for the baseline, LB and MB simulations, modelled $N_{tot}$ are not equal (Fig. 11). The difference is entirely attributable to the dissimilar effectiveness of aerosol processes with lower (LB) and higher (MB) level of detail in representing the aerosol size distribution. Interestingly, using less size bins (LB) has a very minor impact on the horizontal field of $N_{tot}$ while more bins (MB) results in $|\Delta N_{tot}| > 5$ %. This is still

10   smaller than $\Delta N_{tot}$ due to deposition.

Comparing the modelled particulate masses is not that straightforward and is thus not represented here. Already the background concentrations and emissions of particulate mass differ between the simulations because the mass size distribution is calculated from the sectional number size distribution which is different for all simulations.



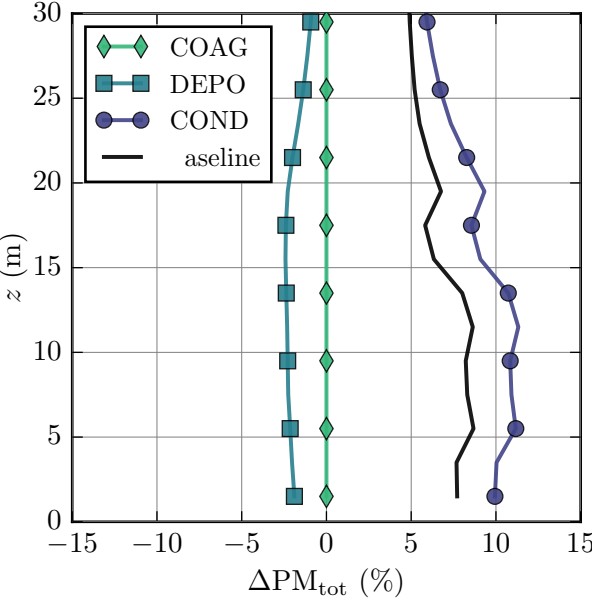

**Figure 10.** Relative difference in particulate mass $\Delta PM_{2.5}$ (%) compared to NOAP for COAG, DEPO, COND and the baseline simulation within the whole evaluation domain in the morning.

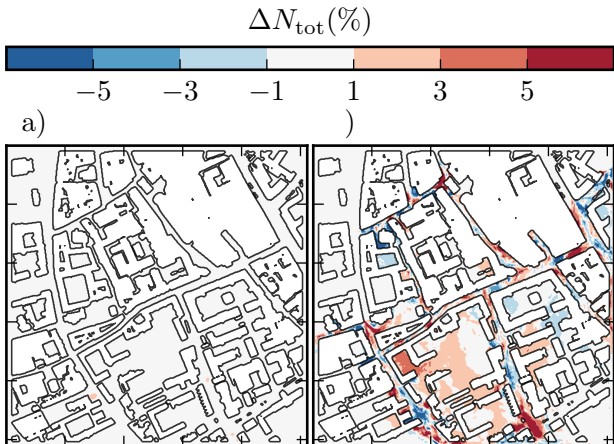

**Figure 11.** Relative difference in the total number concentration $\Delta N_{tot}$ (%) at $z = 3.5$ m compared to the baseline simulation for the a) LB and b) MB simulation in the morning.





## 5 Discussion and conclusions

This article represents a novel, high-resolution, LES-based urban aerosol model that resolves the aerosol particle concentrations, size distributions and chemical compositions at spatial and temporal scales of $1.0\,\mathrm{m}$ and $1.0\,\mathrm{s}$ for entire neighbourhoods.

An evaluation study shows good agreement against measurements on the vertical variation of the aerosol number size
distribution and total number concentration in a simple street canyon in central Cambridge, UK. The model can predict the dilution of concentrations in vertical as well as the number of aerosol particles in different size bins well. The spatial distribution of aerosol concentrations is mostly determined by the flow and emissions. When the individual impact of aerosol dynamic processes, dry deposition is shown to decrease local number concentrations by over 20 %, which is, nonetheless, at the lower end of $\Delta N_{\mathrm{tot}} = [-35, -15]$ % estimated by Huang et al. (2014) for an open space with traffic. Coagulation, instead, has a very
minor impact, which agrees with previous timescale analyses (Kumar et al., 2009; Zhang et al., 2004) and CFD modelling studies (Albriet et al., 2010; Huang et al., 2014; Wang and Zhang, 2012). Condensation and dissolutional growth increase particulate mass by over 10 %. The role of aerosol dynamic processes is shown important both on number and mass especially in areas with low wind speeds, such as in the courtyards and shelters of trees. Furthermore, comparing eight adjacent modelling profiles to one measurement profile shows the limited representativeness of point measurements and gives more support on
performing air quality modelling which gives also the spatial variability of concentrations.

With an increasing modelling complexity, the number of potential sources of modelling uncertainty augments. One of the largest source of uncertainty is related to the quality of the emission data. One major reason to evaluate the aerosol model against the dataset by Kumar et al. (2008) was that mainly only emissions from Pembroke street affected the measured concentrations, which simplified the emission estimated. Modelling uncertainties in the aerosol model cased by simplifying as-
sumptions and model design are discussed in detail in Kokkola et al. (2008). One of the main challenges in simulating both the aerosol number and mass also in this study is the number of aerosol size bins, whereasthe aerosol dynamic processes have less impact. Another inevitable error in sectional aerosol modelling is made when assuming a spherical particle shape and defining the aerosol volume from the bin mean diameter. Despite these limitations, the model simulated the observed concentrations correctly.

Further arguments for applying the selected dataset were, in the first place, the availability of measurements on the vertical variability of aerosol number size distribution at high temporal resolution, but also the simplicity of the urban morphology at the measurement location. The influence of aerosol dynamic processes on aerosol concentration is determined by their size distribution, and thus measurements only on the total number concentration or particulate mass (e.g. Weber et al., 2006) were considered insufficient for this model evaluation. To our knowledge, there exist only a few datasets on the vertical variation
of the aerosol size distribution in an urban environment (Kumar et al., 2008; Li et al., 2007; Marini et al., 2015; Quang et al., 2012; Sajani et al., 2018). Of these datasets, the measurement location of Kumar et al. (2008) in a street canyon with no urban vegetation was simple enough for the first evaluation study. Modelling individual street trees and their aerodynamic impact without exact information on the distribution of leaf area introduces another source of uncertainty for resolving the flow. Furthermore, dry deposition is strongly tree species dependent (e.g. Popek et al., 2013; Sæbø et al., 2012) and therefore





sensitive to correct modelling of different species. Finally, high-resolution topography and land use information were freely available for this specific site.

At the same time, no evaluation data for the flow were available, and therefore the modelling set-up was kept as simple as possible. Hence, the thermal and vehicle induced turbulence were excluded from the simulations. The increase in $N_{\mathrm{tot}}$ between

$z = 1.0 - 2.25\,\mathrm{m}$ observed in the measurements could be explained by any of the two sources of turbulence. Kumar et al. (2008) argued that the increase is likely due to the more efficient dry deposition near the surface or complex dispersion pattern within the canyon caused both by topography and vehicle induced turbulence.

Keeping in mind the aforementioned uncertainties, the presented model provides a novel and flexible tool to study, for example, how the shape, size and location of urban obstacles affect air pollutant transport and transformation. For instance,

the potential of urban vegetation to improve air quality by acting as biological aerosol filters (Beckett et al., 1998) depends on the size-dependent deposition velocity of aerosol particles which is explicitly calculated within the model. The model can also provide information at high enough resolution to perform air pollutant exposure studies or to design a most representative air pollution monitoring network. The aerosol module SALSA can be further coupled with an online chemistry module which are both embedded in the PALM model system as so-called PALM-4U components. This will extent the applicability of

the model from aerosol processes to more complex chemical processes and will allow to examine different urban processes simultaneously such as radiation or thermal comfort.

*Code and data availability.* The PALM code including the sectional aerosol model SALSA can be freely downloaded from http://palm. muk.uni-hannover.de. The distribution is under the GNU General Puclic License v3. More about the code management, versioning and revision control of PALM can be found in Maronga et al. (2015). The standalone version of the SALSA model is freely available at https:

//github.com/UCLALES-SALSA/SALSA-standalone/ and the input datasets at https://doi.org/10.5281/zenodo.1565752 (Kurppa, 2018).

*Author contributions.* MK developed the model code with the support from HK, JT and BM. MK and CK prepared the morphological data and PK the evaluation data. MK, AH, MA and LJ designed the simulations and MK carried them out. MK prepared the manuscript with contributions from all co-authors.

*Competing interests.* No competing interests are present.

*Acknowledgements.* MK acknowledges Sasu Karttunen for technical support, and Basit Khan, Farah Kanani-Sühring, Renate Forkel and Sabine Banzhaf for co-operation, valuable discussions and model testing. This study was financially supported by the Doctoral Programme in Atmospheric Sciences (ATM-DP, University of Helsinki), Helsinki Metropolitan Region Urban Research Program and the Academy of Finland (181255, 277664).





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
