# Peer review of "Implementation of the sectional aerosol module SALSA into the PALM model system 6.0: Model development and first evaluation"

_Geoscientific Model Development, 2018_

## Short Comment (SC1) · 5 Dec 2018

Dear authors, in my role as Executive editor of GMD, I would like to bring to your attention our Editorial version 1.1: http://www.geosci-model-dev.net/8/3487/2015/gmd-8-3487-2015.html This highlights some requirements of papers published in GMD, which is also available on the GMD website in the 'Manuscript Types' section: http://www.geoscientific-model-development.net/submission/manuscript_types.html

In particular, please note that the requirement "The main paper must give the model name and version number (or other unique identifier) in the title." has not been fully met in the Discussions paper: Please provide the version number of SALSA in the title

of your revised manuscript.

Additionally, please note, that GMD is strongly encouraging authors to provide a persistent access to the exact version of the source code used for the model version presented in the paper. As explained in https://www.geoscientific-model-development.net/about/manuscript_types.html the preferred reference to this release is through the use of a DOI which then can be cited in the paper. For projects in GitHub (such as PIC) a DOI for a released code version can easily be created using Zenodo, see https://guides.github.com/activities/citable-code/ for details.

Yours, Astrid Kerkweg

───────────────────────────

---

## Referee Comment (RC1) · Yang (Referee) · 24 Dec 2018

The simulation is very impressive. I have few comments and suggestions the authors may consider.

Page 2, Line 20 to 21.\ We also have few CTAG papers considering the NO-O3-NO2 chemistry. More information could be found from:

Yang, B., Zhang, K.M., Xu, W.D., Zhang, S., Batterman, S., Baldauf, R.W., Deshmukh, P., Snow, R., Wu, Y., Zhang, Q. and Li, Z., 2018. On-Road Chemical Transformation as an Important Mechanism of NO2 Formation. Environmental science & technology,

52(8), pp.4574-4582.

Yang, B. and Zhang, K.M., 2017. CFD-based turbulent reactive flow simulations of power plant plumes. Atmospheric Environment, 150, pp.77-86.

Page 10, line 7-9.\ References needed to show the aerosol dynamic processes are longer than the dispersion process.

Page 13, line 3-5.\ How to estimate LAD = 0.6 mˆ2 mˆ-3?

Page 11, Line 1 and Table 4 caption.\ A quantified wind direction in degrees would be better than the word "northwest". Is that a typo in Table 4 caption, "northeast" ?

Page 14, Line 10 – 11.\ "Horizontal mean U = 40 m" The gradient profile is important to the simulation, so it will be better to provide the velocity profile.

Page 14, Line 16.\ References needed for the roughness height and drag coefficient of trees.

Figure 5.\ Scatter points would be more appropriate for showing the measurement data because they were at four different heights above the ground level. In addition, a local plot from the ground level to 10 m would be good enough for this plot. The x-axis can also be enlarged because of the log scale.

---

## Referee Comment (RC2) · Anonymous Referee #2 · 27 Jan 2019

General comments:

This paper presents a novel modelling tool that couples a sectional aerosol module with a CFD based atmospheric driver. The model is used to simulate the measurements on the vertical variation of the aerosol number size distribution and concentrations in a street-canyon (Pembroke Street) in central Cambridge, United Kingdom, on March 20–21, 2007 (Kumar et al., 2008, 2009). The model represents the state-of-the-art in micro-scale air quality simulations and the results presented are of interest for the urban air quality research community. The only point that should be furthermore discussed by the authors are the limitations of CFD and LES approaches over neighbour-

hood and city scale domains due to its large computational resources required for a single hour simulation. The authors should clarify what is currently the range of applications of this type of model. Its complexity doesn't make them suitable for the study of the urban scale air pollution if large computational resources are not available. The authors also state that emissions are a critical part to achieve realistic model results of size distribution. A discussion of the advantages and limitations of this approach should be included in the introduction.

Overall, the manuscript is well written and presents a novel approach to model size distribution of aerosols at urban scales. In my opinion, this paper deserves a minor revision to be published in Geoscientific Model Development. I recommend the authors to address the general comment and improve the manuscript following the specific and technical comments detailed below.

Specific comments:

- Page 1, line 6: the authors use qualitative comments to present the skills of the model like "excellent agreement with measurements" through all the text. The manuscript will be improved if such statements are accompanied by quantitative statistics.

- Page 2, line 6: please, add a reference to the sentence about Gaussian dispersion or semi-empirical street models limitations to represent urban complexity. More than the representation of urban complexity, such models have major limitations in the dispersion and to represent fine-scale flow processes.

- Page 3, line 3: it sounds contradictory to select an aerosol module initially designed for large scale applications. The authors should clarify the requirements needed for high-resolution micro-scale simulations and better justify the use of SALSA in the model approach presented.

- Page 3, line 6: the authors mention that the model is evaluated under different meteorological conditions, but in the manuscript, only neutral meteorological conditions are

modeled. Please, correct this statement.

- Page 3, line 14: what "filtered" means in this framework?

- Page 3, line 23: the authors should quantify the sentence "Due to its excellent scalability on massively parallel computer architectures" or provide a reference.

- Page 3, line 29: what are the aspects that mostly affect computational expenses in aerosol microphysical modules? SALSA uses a sectional approach, which is not the best option to limit computational expenses in a micro-scale model.

- Page 3, line 30: the authors should justify the sentence "SALSA is equally suitable for presenting aerosol dynamics also at local scale". What are the requirements on aerosol processes in local scale models compared with large-scale models?

- Page 4, line 2: Are all the aerosols represented with 10 sectional bins?

- Page 4, line 3 and 4: Please, clarify that sulphuric acid, nitric acid, and ammonia can condense in the particle phase, and condensation is the only process forming sulphate, nitrate, and ammonium in the model.

- Page 4, line 17: Is the resuspension of coarse aerosols considered in the model? The authors should comment on the implications of excluding this process. Resuspension of particles by vehicles is an important emission source at the urban scale.

- Page 9, caption Table 2: the base case run should be included in the table to better understand the relative change in the total computation.

- Page 11, line 8: the authors have to include in Figure 3 or in a new figure the location of the measurements and indicate which are used to compare with the model.

- Page 12, caption Figure 3: were the observations taken at the same location as the red crosses? It is not clear in the manuscript where was the location of the observations.

- Page 14, line 4: how are the background profiles of aerosols ingested in the model and which size distribution is assumed? This may have a significant impact on the results of the model.

- Page 14, line 14: how is the turbulent kinetic energy initialized in the model? This is one of the critical points of LES models.

- Page 15, line 23: The authors mention that the modelled and measured values are hourly averaged. However, it would be quite interesting to see the histogram of measurements and simulation results during that hour. The high-resolution capability of the current modelling tool allows this analysis and it would provide interesting information about the skills of the model to reproduce the micro-scale features identified in the observations.

- Page 16, line 1: "compare well", the authors should quantify this statement. The scale of Figure 5 is semi-logarithmic which makes the differences with observations difficult to appreciate. Model results may be a factor of 2 or more overestimated compared with measurements. The authors should put in perspective this result. Are those differences reasonable when modelling number concentration?

- Page 16, line 2: "slight overestimation", the authors should quantify this.

- Page 17, line 1: "The rate of change of Ntot in vertical is correctly modelled except for a measured increase in concentrations within the lowest 2m". Can the authors discuss this? Is there a specific process that may explain this difference with the observations? Dry deposition would be the first process to consider.

- Page 17, Figure 5: It would be useful to plot the sigma of the observations together with the mean for the hourly average. This information would provide an idea of the variability within one hour observed in the area of study.

- Page 20, line 9: "there are no traffic related emissions of gaseous HNO3". Why this has not been considered in the simulation? NOx emissions are important in traffic,

and they are a source for the formation of HNO3. Its inclusion in the simulation may significantly increase the condensation of HNO3 on the particles.

- Page 21, line 11: although the comparison of particulate mass could be not straight-forward, it would provide some initial guidance on how uncertainty in number concentration affects mass concentration. A comparison of the model with the mass concentration of the observations would be quite interesting and an added value for the manuscript.

- Page 23, line 4: "good agreement", please quantify.

- Page 24, line 3: "no evaluation data were available". In section 3.4 and Table 4, the authors mention that the wind at 40m was adjusted to the observed one. This suggests that some meteorological data were available from the measurement campaign. Why this information is not used to evaluate the wind of the model?

Technical comments:

- Figures: all the figures have errors in the legend or letters used to identify the different panels. Please, make a complete revision of all of them and fix the problems with legends and letters.

- Page 1, title: please, specify the version of the module SALSA implemented.

- Page 2, line 25: correct "as an superposition" with "as a superposition".

- Page 3, line 7: correct "aerosol size distributions and chemical compositions" with "aerosol size distribution and chemical composition".

- Page 3, line 13: correct "an LES core" with "a LES core".

- Page 4, Table 1: correct "Is a surface scheme is switched on" with "If the surface scheme is switched on".

- Page 4, line 6: correct "Nitrates and ammonia" with "Nitrates and ammonium".

- Page 5, equation 2: define "vc" in the text.

- Page 8, equation 10: define "LAD" in the text.

- Page 12, Figure 3: identify Pembroke Street in the map.

- Page 20: Figure 8 is not explicitly mentioned in the text, please do so or remove the figure from the manuscript.

- Page 22, caption Figure 10: correct "PM2.5" with "PMtot".

- Page 23, line 19: correct "model cased by" with "model caused by".

- Page 23, line 21: correct "whereasthe" with "whereas the".

- Page 23, line 23: correct "observed concentrations" with "observed number concentrations".

---

## Author Comment (AC1) · 22 Feb 2019

Dear Executive Editor,

Thank you for your comments. We have now provided the version number of SALSA (2.0) in the title and throughout the manuscript. Additionally, the exact version of the source code is now available in 10.5281/zenodo.2575325.

---

## Author Comment (AC2) · 22 Feb 2019

Dear Bo Yang,

Please find as a supplement our point-by-point response to the Referee comments and also a file showing the changes made in the manuscript.

Please also note the supplement to this comment:
https://www.geosci-model-dev-discuss.net/gmd-2018-282/gmd-2018-282-AC2-supplement.zip

---

## Author Response (AR1)

**Referee response to manuscript Implementation of the sectional aerosol module SALSA into the PALM model system 6.0: Model development and first evaluation**

We thank both reviewers for their valuable comments and suggestions. Please find our detailed point-by-point responses below (in black).

The changes made to the manuscript are visualised in the attached file "manuscript\_see\_differences.pdf". Page and line numbers given in this response refer to that document.

**Referee #1 (Yang):**

The simulation is very impressive. I have few comments and suggestions the authors may consider.

Page 2, Line 20 to 21.\ We also have few CTAG papers considering the NO-O3-NO2 chemistry. More information could be found from:

Yang, B., Zhang, K.M., Xu, W.D., Zhang, S., Batterman, S., Baldauf, R.W., Deshmukh, P., Snow, R., Wu, Y., Zhang, Q. and Li, Z., 2018. On-Road Chemical Transformation as an Important Mechanism of NO2 Formation. Environmental science & technology, 52(8), pp.4574-4582.

Yang, B. and Zhang, K.M., 2017. CFD-based turbulent reactive flow simulations of power plant plumes. Atmospheric Environment, 150, pp.77-86.

We thank you for bringing up these research articles! Here, we however focus on discussing only simulations including aerosol particles and not the gas phase. To clarify this, we modified the phrase slightly:

"The CTAG model has also been run in a LES mode (Steffens et al., 2013), but to date aerosol simulations have only considered dry deposition (Tong et al., 2016a, b)..." (P2 L27-29).

Page 10, line 7-9.\ References needed to show the aerosol dynamic processes are longer than the dispersion process.

A reference added to Pryor and Binkowski (2004) and Kumar et al. (2008) (P11 L2)

**Page 13, line 3-5.\ How to estimate LAD = $0.6 \text{ m}^2 \text{ m}^-3$ ?**

A phrase is added:

"This LAD value was estimated as a lower limit for urban street trees in Northern Europe in spring (Gillner et al., 2015)" (P14 L2-3)

Page 11, Line 1 and Table 4 caption.\ A quantified wind direction in degrees would be better than the word "northwest". Is that a typo in Table 4 caption, "northeast" ?

Yes, you are correct. Thank you for noting this typo!

Page 14, Line 10 - 11. "Horizontal mean U = 40 m" The gradient profile is important to the simulation, so it will be better to provide the velocity profile.

We have now included the 1-hour mean vertical profiles of horizontal wind and Reynolds stress to the supplementary material. (P15 L9-10, Fig. S7)

Page 14, Line 16.\ References needed for the roughness height and drag coefficient of trees.

References added both for the roughness length (Letzel et al., 2012) and drag coefficient of trees (Kent et al., 2017). (P15 L15-16)

Figure 5.\ Scatter points would be more appropriate for showing the measurement data because they were at four different heights above the ground level. In addition, a local plot from the ground level to 10 m would be good enough for this plot. The x-axis can also be enlarged because of the log scale.

The 1-hour averaged measurements are now shown with scatter points. Furthermore, additional linear plots zooming into the lowest 10 m are added to Figure 5.

**Referee #2 (Anonymous):**

**General comments:**

This paper presents a novel modelling tool that couples a sectional aerosol module with a CFD based atmospheric driver. The model is used to simulate the measurements on the vertical variation of the aerosol number size distribution and concentrations in a street-canyon (Pembroke Street) in central Cambridge, United Kingdom, on March 20–21, 2007 (Kumar et al., 2008, 2009). The model represents the state-of-the-art in micro-scale air quality simulations and the results presented are of interest for the urban air quality research community. The only point that should be furthermore discussed by the authors are the limitations of CFD and LES approaches over neighbourhood and city scale domains due to its large computational resources required for a single hour simulation. The authors should clarify what is currently the range of applications of this type of model. Its complexity doesn't make them suitable for the study of the urban scale air pollution if large computational resources are not available. The authors also state that emissions are a critical part to achieve realistic model results of size distribution. A discussion of the advantages and limitations of this approach should be included in the introduction.

Overall, the manuscript is well written and presents a novel approach to model size distribution of aerosols at urban scales. In my opinion, this paper deserves a minor revision to be published in Geoscientific Model Development. I recommend the authors to address the general comment and improve the manuscript following the specific and technical comments detailed below.

Thank you for this valuable comment concerning the computational limitations of LES approach in neighbourhood to city scale air quality modelling and also the importance of accurate emission estimates. We have now added the following discussion to the manuscript:

"Fortunately, constantly increasing computational power has already allowed urban LES modelling for entire neighbourhoods up to one day or even more in a supercomputing environment (e.g. Resler et al., 2017)." (P2 L22-23)

"In any case, the computational expenses are multiplied when SALSA is included, which limits the size of LES model domains to be considered. " (P11 L3-4)

"Keeping in mind the aforementioned uncertainties **and required computational resources**, the presented model provides a novel and flexible tool to study, for example, how the shape, size and location of urban obstacles affect air pollutant transport and transformation **at a neighbourhood scale**." (P25 L20-22)

"Moreover, ongoing model development aims at extending the application of the model from supercomputing environments to personal PCs in future (Maronga et al., 2019)." (P25 L28-30)

Furthermore, a paragraph discussing the advantages and limitations of the need for accurate emission data was added to the introduction:

"The fate of aerosol particles in the atmosphere depends substantially on their size distribution. Consequently, detailed aerosol modelling requires size-specific emission and background information as input. Estimates for background aerosol size distributions and concentrations can be attained from larger scale models, whereas emission data is usually treated as total aerosol mass. Hence, emission size distribution has to be estimated based on the source type and vehicle fleet in case of traffic emissions. If any important emission source is neglected, aerosol processes are also calculated erroneously. At the same time, as LES outperforms traditionally used urban air quality models in resolving the turbulent find field and pollutant dispersion, LES-based air quality models produce unique information on pollutant transformation and dispersion processes with accurate emission estimates." (P2 L30-34 & P3 L1-3)

**Specific comments:**

- Page 1, line 6: the authors use qualitative comments to present the skills of the model like "excellent agreement with measurements" through all the text. The manuscript will be improved if such statements are accompanied by quantitative statistics.

We have now included quantitative statistics (fractional bias FB and factor of two FAC2) to evaluate the model performance (see Figures S9 and S10 in SI).

This specific phrase was rewritten as follows:

"The first model evaluation study on the vertical variation of aerosol number concentration and size distribution in a simple street canyon without vegetation in Cambridge, UK, shows **good agreement with measurements with simulated values mainly within a factor of two of observations.**" (P1 L5-7) - Page 2, line 6: please, add a reference to the sentence about Gaussian dispersion or semi-empirical street models limitations to represent urban complexity. More than the representation of urban complexity, such models have major limitations in the dispersion and to represent fine-scale flow processes.

**We added a phrase**

"... and limitations in resolving any fine-scale flow structures "

with a reference to Tominaga and Stathopoulos (2016) which nicely summarizes the benefits and limitations of both (semi-)empirical and Gaussian dispersion models in urban air quality studies. (P2 L10-11)

- Page 3, line 3: it sounds contradictory to select an aerosol module initially designed for large scale applications. The authors should clarify the requirements needed for high-resolution micro-scale simulations and better justify the use of SALSA in the model approach presented.

Although micro-scale simulations and global scale climate models simulate quite different scale domains, both types of models require extreme computational efficiency for the aerosol microphysics calculations. In both types of models, the computational burden comes from calculating aerosol microphysics in very large number of grid points. Thus, the design choices in SALSA, which aim at optimizing the balance between the computational efficiency and numerical accuracy of calculating aerosol microphysics in global scale models, are also valid for micro-scale simulations. On the other hand, extending SALSA to include the partitioning of ammonia and nitric acid between gas and particle phase makes it more suitable for air quality simulations in LES framework while making it computationally very demanding for global scale simulations, especially for long simulation periods. This is now clarified in the revised manuscript:

"SALSA2.0 (referred to hereafter simply as SALSA) was selected as the basis for representing aerosol dynamics in PALM since one major criteria in its development has been limiting computational expenses without the cost of accuracy. A major share of the expenses stem from having a large number of prognostic variables to describe the aerosol population. SALSA has been optimized for resolving aerosol microphysics in very large number of grid points, such as in global-scale climate models, but the same processes are relevant also at local scale. Nonetheless, the same aerosol processes and model design choices are valid at local scale." (P4 L14-16 & P5 L1-3)

- Page 3, line 6: the authors mention that the model is evaluated under different meteorological conditions, but in the manuscript, only neutral meteorological conditions are modeled. Please, correct this statement.

We changed the word "meteorological" to "wind" (P3 L22)

- Page 3, line 14: what "filtered" means in this framework?

Filtering refers here to low-pass filtering in LES, i.e. the Navier-Stokes equations are filtered or separated into the explicitly calculated resolved scales and sub-grid scales which are parametrised.

- Page 3, line 23: the authors should quantify the sentence "Due to its excellent scalability on massively parallel computer architectures" or provide a reference.

A sentence: "(*up to 50,000 processor cores; Maronga et al. 2015*)" was added. (P4 L8)

- Page 3, line 29: what are the aspects that mostly affect computational expenses in aerosol microphysical modules? SALSA uses a sectional approach, which is not the best option to limit computational expenses in a micro-scale model.

The costs usually stem from having a large number of prognostic variables, i.e. here variables to describe the aerosol size distribution. On the other hand, the sectional method can describe the aerosol dynamic processes more accurately (see e.g. Zhang et al., Impact of aerosol size representation on modeling aerosol-cloud interactions, *J. Geophys. Res.*, 107(D21), 4558, doi:10.1029/2001JD001549, 2002).

The following phrase was added:

"A major share of the expenses stem from having a large number of prognostic variables to describe the aerosol population." (P4 L16 & P5 L1)

- Page 3, line 30: the authors should justify the sentence "SALSA is equally suitable for presenting aerosol dynamics also at local scale". What are the requirements on aerosol processes in local scale models compared with large-scale models?

As explained before, both types of models require that the aerosol processes are calculated computationally efficiently in large number of grid points, while maintaining numerical accuracy. The same aerosol microphysical processes, i.e. nucleation, condensation/evaporation, coagulation, and dissolutional growth, affect the aerosol properties that are relevant for global scale climate simulations, global chemistry transport models simulations, and local scale air quality simulations.

**- Page 4, line 2: Are all the aerosols represented with 10 sectional bins?**

The default number of aerosol size bins is 10.

- Page 4, line 3 and 4: Please, clarify that sulphuric acid, nitric acid, and ammonia can condense in the particle phase, and condensation is the only process forming sulphate, nitrate, and ammonium in the model.

Sulphuric acid condensates directly on aerosol particles while nitric acid and ammonia dissolve in liquid water on the aerosol particles. Traffic emissions introduce gaseous sulphuric acid and ammonia as well as sulphates in the particle phase to the model domain.

The following phrase was added for clarification:

"Furthermore, the gaseous concentrations of H2SO4, HNO3, NH3 and semi- and non volatile organics (SVOC and NVOC), **that can condense or dissolve on aerosol** *particles*, are also default prognostic variables." (P5 L9-10)

- Page 4, line 17: Is the resuspension of coarse aerosols considered in the model? The authors should comment on the implications of excluding this process. Resuspension of particles by vehicles is an important emission source at the urban scale.

The phrase was modified to:

"The process of particle resuspension from surfaces is currently neglected. However, resuspension of road dust, for example, can be included in the model as an additional surface emission (see Section 2.2.5)" (P5 L22-24)

- Page 9, caption Table 2: the base case run should be included in the table to better understand the relative change in the total computation.

The computational time is hardware specific and therefore we think is not relevant to provide the exact computational time of the base run.

- Page 11, line 8: the authors have to include in Figure 3 or in a new figure the location of the measurements and indicate which are used to compare with the model.

The exact measurement location (66 m from the intersection in the southwest) is now marked on the Figure 3.

- Page 12, caption Figure 3: were the observations taken at the same location as the red crosses? It is not clear in the manuscript where was the location of the observations.

The exact location is now marked to the Figure 3 with a black cross. The red crosses indicate additional profiles where the simulated concentrations were compared against the measured one.

- Page 14, line 4: how are the background profiles of aerosols ingested in the model and which size distribution is assumed? This may have a significant impact on the results of the model.

The background aerosol concentrations and size distribution are produced by the trajectory model ADCHEM (see P14 L22-27 & P15 1-4; and Section S5 in SI) for the specific simulation times. The background is introduced to the simulation domain by a decycling method, in which the constant background concentrations are fixed at the lateral boundaries.

- Page 14, line 14: how is the turbulent kinetic energy initialized in the model? This is one of the critical points of LES models.

Unfortunately we do not entirely understand this question.

The model simulations are initialised with arbitrary logarithmic wind profiles and the flow is forced with a pressure gradient in the layer above z = 120 m. When the resolved wind field becomes stable, the solution no longer depends on the initial state of the simulation. The

most important is that turbulence is formed in the first place and that the model resolution is high enough so that most of the kinetic energy is directly resolved and not parametrised.

- Page 15, line 23: The authors mention that the modelled and measured values are hourly averaged. However, it would be quite interesting to see the histogram of measurements and simulation results during that hour. The high-resolution capability of the current modelling tool allows this analysis and it would provide interesting information about the skills of the model to reproduce the micro-scale features identified in the observations.

We really agree with the comment. Unfortunately, only hourly averages of measurements were available for the simulation time. Here we want to emphasize that only a few measurement campaigns (Kumar et al., 2008; Li et al., 2007; Marini et al., 2015; Quang et al., 2012; Sajani et al., 2018) on the vertical variation of aerosol size distribution in urban environment have been conducted. Of these, the measurements by Kumar et al. (2008) are the only ones for which information on the topography and pollutant emissions are freely available and where there was no vegetation close to the measurement site (as we wanted to keep the evaluation environment as simple as possible and minimize any modelling error caused by modelling vegetation in LES). Thus, the used dataset is suitable only for the current study.

Instead of showing modelled concentrations only at one horizontal point, we wanted to include the spatial variability as well. Therefore including temporal variability of simulations would have made the figures chaotic.

- Page 16, line 1: "compare well", the authors should quantify this statement. The scale of Figure 5 is semi-logarithmic which makes the differences with observations difficult to appreciate. Model results may be a factor of 2 or more overestimated compared with measurements. The authors should put in perspective this result. Are those differences reasonable when modelling number concentration?

As mentioned before, we have now included quantitative statistical analysis to the manuscript. Please see Figures S9 and S10 in SI. Additionally, the following phrases were included:

"The modelled mean vertical profiles of  $N_{tot}$  compare well against the measured values (Fig. 5) especially in the morning. Indeed, also the additional six profiles are generally within the factor of two of observations (see Fig. S9 in SI)." (P17 L3-4 & P18 L1)

Additional linear plots zooming into the lowest 10 m are added to Figure 5. Furthermore, the model error related to the modelled values of aerosol number concentrations is discussed as follows:

"This deviation from measurements is comparable to typical differences in measured aerosol number concentrations with different instruments (Ankilov et al., 2002; Hornsby and Pryor, 2014)." (P18 L4-5)

- Page 16, line 2: "slight overestimation", the authors should quantify this.

The phrase was modified to:

"Despite the modelled mean  $N_{tot}$  being 50-100 % higher than the measured in the evening..." (P18 L2-3)

- Page 17, line 1: "The rate of change of Ntot in vertical is correctly modelled except for a measured increase in concentrations within the lowest 2m". Can the authors discuss this? Is there a specific process that may explain this difference with the observations? Dry deposition would be the first process to consider.

This is discussed in the Discussion section:

"... Hence, the thermal and vehicle induced turbulence were excluded from the simulations. The increase in  $N_{tot}$  between z = 1.0-2.25 m observed in the measurements could be explained by any of the two sources of turbulence. Kumar et al. (2008) argued that the increase is likely due to the more efficient dry deposition near the surface or complex dispersion pattern within the canyon caused both by topography and vehicle induced turbulence." (P25 L16-19)

- Page 17, Figure 5: It would be useful to plot the sigma of the observations together with the mean for the hourly average. This information would provide an idea of the variability within one hour observed in the area of study.

As explained above, only hourly averages of observations were unfortunately available.

- Page 20, line 9: "there are no traffic related emissions of gaseous HNO3". Why this has not been considered in the simulation? NOx emissions are important in traffic, and they are a source for the formation of HNO3. Its inclusion in the simulation may significantly increase the condensation of HNO3 on the particles.

When the simulations were conducted, the chemistry model included in PALM was not ready to be applied yet. Therefore, production of HNO3 by chemical reactions was not included. We agree that included the NOx chemistry can change the results. However, the direct emissions of HNO3 from traffic are very low and hence neglected here.

- Page 21, line 11: although the comparison of particulate mass could be not straightforward, it would provide some initial guidance on how uncertainty in number concentration affects mass concentration. A comparison of the model with the mass concentration of the observations would be quite interesting and an added value for the manuscript.

In this study, we primarily wanted to focus on the size distribution of aerosol particles and different aerosol processes. As mentioned before, suitable size distribution datasets are rare and unfortunately, the one we used from Cambridge includes measurements on the aerosol number size distribution only. Therefore, we could not evaluate the model performance in modelling aerosol mass concentrations. This is, however, planned next step in future studies.

- Page 23, line 4: "good agreement", please quantify.

Quantitative estimate was given and the phrase was modified as follows:

"An evaluation study on the vertical variation of the aerosol number size distribution and total number concentration in a simple street canyon in central Cambridge, UK, shows good agreement against measurements. The model can predict the dilution of concentrations in vertical as well as the number of aerosol particles in different size bins generally within a factor of two of observations." (P23 L14-15 & P24 L1-2)

- Page 24, line 3: "no evaluation data were available". In section 3.4 and Table 4, the authors mention that the wind at 40m was adjusted to the observed one. This suggests that some meteorological data were available from the measurement campaign. Why this information is not used to evaluate the wind of the model?

Here we mean that there were no temporally nor spatially high-resolution measurements on wind speed and direction available to evaluate the performance of PALM on resolving the wind field but simply 10-min averaged synoptic observations. We modified the sentence to:

"At the same time, no **high-resolution** evaluation data for the flow were available, and therefore the modelling set-up was kept as simple as possible." (P25 L15-16)

**Technical comments:**

- Figures: all the figures have errors in the legend or letters used to identify the different panels. Please, make a complete revision of all of them and fix the problems with legends and letters.

We apologize for some bug that occurred when creating the pdf-file, which messed up the texts in all figures. This will be fixed in the revised paper.

- Page 1, title: please, specify the version of the module SALSA implemented.

The version number of SALSA (2.0) was added to the title and also to the main text.

- Page 2, line 25: correct "as an superposition" with "as a superposition".

"an" was replaced with "a" (P3 L6)

- Page 3, line 7: correct "aerosol size distributions and chemical compositions" with "aerosol size distribution and chemical composition".

The plural nouns were replaced by singulars (P3 L23-24)

- Page 3, line 13: correct "an LES core" with "a LES core".

The pronunciation of LES starts by a vowel sound "*el*" and thus, according to my knowledge of English grammar, an LES should be the correct form.

- Page 4, Table 1: correct "Is a surface scheme is switched on" with "If the surface scheme is switched on".

Thank you noticing the typo. The phrase was corrected accordingly.

- Page 4, line 6: correct "Nitrates and ammonia" with "Nitrates and ammonium".

"ammonia" was replaced with "ammonium" (P5 L11)

- Page 5, equation 2: define "vc" in the text.

A definition was added:

" $v_{c,i}$  is the aerosol volume concentration of chemical component c in size bin i and  $\rho_c$  is its density" (P6 L4)

- Page 8, equation 10: define "LAD" in the text.

A definition was added:

"which depends on the local leaf area density (LAD) ..." (P8 L13)

- Page 20: Figure 8 is not explicitly mentioned in the text, please do so or remove the figure from the manuscript.

Thank you for noticing this. We were supposed to refer to Fig. 8. This was now corrected:

"Neglecting all aerosol processes overestimates  $N_{tot}$  (see Fig. S11 in SI), and therefore including dry deposition is essential for modelling realistic  $N_{tot}$ ." (P19 L20-21)

- Page 22, caption Figure 10: correct "PM2.5" with "PMtot".

"PM2.5" was replaced with "PMtot". Thanks!

- Page 23, line 19: correct "model cased by" with "model caused by".

The phrase was reformatted.

- Page 23, line 21: correct "whereasthe" with "whereas the".

"whereasthe" was replaced with "whereas the" (P24 L19)

- Page 23, line 23: correct "observed concentrations" with "observed number concentrations".

"number" was added (P25 L2)

**Implementation of the sectional aerosol module SALSA SALSA2.0 into the PALM model system 6.0: Model development and first evaluation**

Mona Kurppa1, Antti Hellsten2, Pontus Roldin1,3, Harri Kokkola4, Juha Tonttila4, Mikko Auvinen1,2, Christoph Kent5, Prashant Kumar6, Björn Maronga7,8, and Leena Järvi1,9 1Institute 
[revised manuscript text omitted]

| 10  | $\mathrm{H}_2\mathrm{SO}_4,\mathrm{OC},\mathrm{HNO}_3,\mathrm{NH}_3$             | 65                  | condensation         | yes                        | $\Delta t$            | + 820                                |
| 11  | $\mathrm{H}_2\mathrm{SO}_4,\mathrm{OC},\mathrm{HNO}_3,\mathrm{NH}_3,\mathrm{BC}$ | 75                  | all                  | yes                        | $\Delta t$            | + 1370                               |
| 12  | $\mathrm{H}_2\mathrm{SO}_4,\mathrm{OC},\mathrm{HNO}_3,\mathrm{NH}_3,\mathrm{BC}$ | 75                  | all                  | yes                        | $2\Delta t$           | + 1130                               |
| 13  | H 2 SO 4 , OC, HNO 3 , NH 3 , BC     | 75                  | all                  | yes                        | $10\Delta t$          | + 810                                |

Each  $n_i$ ,  $m_{c,i}$  and gaseous compound introduces a new prognostic variable that is transported by the flow in PALM. Increasing the number of prognostic variables  $X_{PV}$  from the default value of  $X_{PV} = 6$  (wind components u, v, w and scalars e,  $\theta$  and q) to

5
$$X_{\rm PV} = 6 + \Delta X_{\rm PV} = 6 + X_{\rm B}(X_{\rm CC} + 1) + X_{\rm G}$$
, (12)

[revised manuscript text omitted]

Aerosol size distributions in the size range D = 5-2738 nm were measured pseudo-simultaneously at four heights (z = 1.00, 2.25, 4.62 and 7.37 m above ground level (AGL)) on the northwestern side of Pembroke street using a fast-response differ-

20 in